# What Can Transformers Learn In-Context?
# A Case Study of Simple Function Classes

**Shivam Garg**[*] **Dimitris Tsipras**[*] **Percy Liang** **Gregory Valiant**

Stanford University
{shivamg, tsipras, pliang, gvaliant}@cs.stanford.edu

## Abstract

In-context learning refers to the ability of a model to condition on a prompt sequence consisting of in-context examples (input-output pairs from some task) along with a new query input, and generate the corresponding output. Crucially, in-context learning happens only at inference time without any parameter updates to the model. While large language models such as GPT-3 exhibit some ability to perform in-context learning, it is unclear what the relationship is between tasks on which this succeeds and what is present in the training data. To make progress towards understanding in-context learning, we consider the well-defined problem of training a model to in-context learn a function class (e.g., linear functions): given data derived from some functions in the class, can we train a model to in-context learn "most" functions from this class? We show empirically that standard Transformers can be trained from scratch to perform in-context learning of linear functions—that is, the trained model is able to learn unseen linear functions from in-context examples with performance comparable to the optimal least squares estimator. In fact, in-context learning is possible even under two forms of distribution shift: (i) between the training data of the model and inference-time prompts; (ii) between the in-context examples and the query input during inference. We also show that we can train Transformers to in-context learn more complex function classes—i.e., sparse linear functions, two-layer neural networks, and decision trees—with performance that matches or exceeds task-specific learning algorithms.[1]

## 1 Introduction

Large language models such as GPT-3 [10] are able to perform *in-context learning*: given a prompt containing examples from a task (input-output pairs) and a new query input, the language model can generate the corresponding output. For example, these models are able to produce English translations of French words after being prompted on a few such translations, e.g.:

$$\underbrace{\text{maison} \rightarrow \text{house, chat} \rightarrow \text{cat, chien} \rightarrow}_{\text{prompt}} \underbrace{\text{dog}}_{\text{completion}} .$$

This capability is quite intriguing as it allows models to adapt to a wide range of downstream tasks on-the-fly—without the need to update the model after training [10, 35, 55, 8]. However, it is unclear to what extent these models are able to learn *new tasks* from in-context examples alone as opposed to indexing into a vast set of known tasks from the training data (e.g., see Min et al. [41]). [2]

---

[*]Equal contribution.

[1]Our code and models are available at `https://github.com/dtsip/in-context-learning`.

[2]The term "in-context learning" may also refer to a more general notion of learning from a prompt [48]. In this work, we focus on the standard notion which refers to learning a task/function given in-context examples [10].

To make progress towards understanding in-context learning, we consider the well-defined problem of learning a *function class* from in-context examples. That is, we say that a model can in-context learn a function class $\mathcal{F}$ if, for "most" functions $f \in \mathcal{F}$, the model can approximate $f(x_{\text{query}})$ for a new query input $x_{\text{query}}$ by conditioning on a prompt sequence $(x_1, f(x_1), \ldots, x_k, f(x_k), x_{\text{query}})$ containing in-context examples and the query input.

Formally, let $D_{\mathcal{X}}$ be a distribution over inputs and $D_{\mathcal{F}}$ be a distribution over functions in $\mathcal{F}$. A prompt $P$ is a sequence $(x_1, f(x_1), \ldots, x_k, f(x_k), x_{\text{query}})$ where inputs (i.e., $x_i$ and $x_{\text{query}}$) are drawn i.i.d. from $D_{\mathcal{X}}$ and $f$ is drawn from $D_{\mathcal{F}}$. We say that a model $M$ can in-context learn the function class $\mathcal{F}$ up to $\epsilon$, with respect to $(D_{\mathcal{F}}, D_{\mathcal{X}})$, if it can predict $f(x_{\text{query}})$ with an average error

$$\mathbb{E}_P\left[\ell\left(M\left(P\right), f\left(x_{\text{query}}\right)\right)\right] \leq \epsilon, \tag{1}$$

where $\ell(\cdot, \cdot)$ is some appropriate loss function, such as the squared error.

Within this framework, we can now concretely ask:

*Can we train a model to in-context learn a certain function class?*

Note that, here, being able to in-context learn a function class is a property of model $M$ alone, independent of how it was trained. *Training* such a model can be viewed as an instance of *meta-learning* [62, 45, 67], a general paradigm for learning a model or method that can learn from data.

We empirically study this question, focusing on Transformer models [69, 53]—the architecture behind recent large language models—trained from scratch to in-context learn simple, well-defined function classes (e.g. linear functions). Specifically, we sample prompts containing in-context examples (input-output pairs) generated using functions in a given class and train models to predict the function value at the corresponding query inputs. (see illustration in Figure 1). Our findings are as follows.

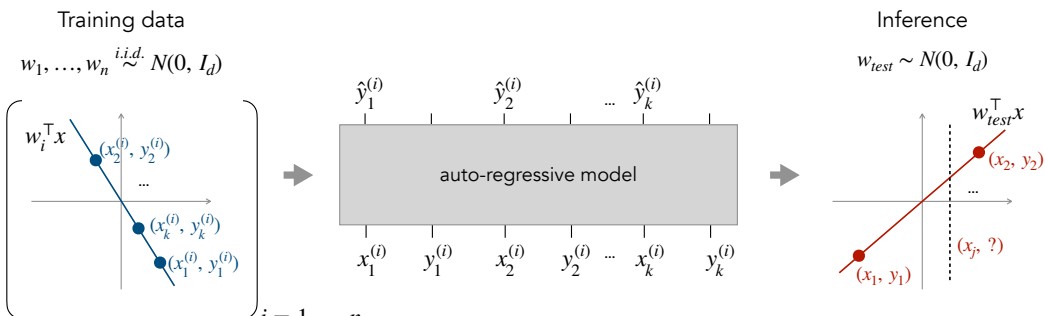

Figure 1: *Can we train a model that in-context learns a function class (here linear functions)? We train Transformers by repeatedly sampling a random function $f$ from that class, as well as random inputs $x_1, \ldots, x_k$ and training the model to predict each $f(x_i)$ given the prompt $x_1, f(x_1), \ldots, x_{i-1}, f(x_{i-1}), x_i$ (wrt squared loss). Then, during inference, we evaluate the model's ability to predict accurately on new, unseen functions.*

**Transformers can in-context learn linear functions.** We show empirically that we can train a standard Transformer from scratch to in-context learn the class of linear functions, with respect to the input distribution $D_{\mathcal{X}}$ being an isotropic Gaussian in 20 dimensions, and $D_{\mathcal{F}}$ being the distribution over linear functions with weight vectors drawn from an isotropic Gaussian (the model was trained on prompts generated from the same distributions $D_{\mathcal{X}}$ and $D_{\mathcal{F}}$). Specifically, the trained model achieves error comparable to the optimal least squares estimator, suggesting that it encodes an effective learning algorithm, at least for the distribution used to generate the training prompts.

**Generalization to out-of-distribution prompts.** To understand the extent to which the trained model encodes an algorithm that works beyond the training distribution, we consider in-context learning under two types of distribution shifts: (a) a shift between the prompts encountered during training and inference (e.g., training on prompts without any noise in the in-context example outputs but testing with noisy outputs), (b) a shift between the in-context examples and the query input during inference (e.g., in-context examples lie in one orthant and the query input lies in another). We find that the performance of our model is quite robust to such shifts, indicating that it has learned to perform linear regression with some generality.

**More complex function classes.** We also consider the function classes of 3-sparse linear functions, two-layer ReLU neural networks with 100 hidden units, and decision trees of depth 4, all with 20 dimensional inputs. We show that we can train Transformer models that can in-context learn these classes (with respect to isotropic Gaussian inputs and appropriately defined distributions over functions). For sparse linear functions, the trained model exploits sparsity, obtaining performance better than least squares and comparable to Lasso. For neural networks, the trained model performs comparably to neural networks of the same architecture trained using gradient descent on in-context examples. Moreover, it is also able to in-context learn linear functions. For decision trees, the trained model learns unseen trees with as few as 100 in-context examples, whereas greedy learning and tree boosting algorithms are unable to achieve competitive performance. Note that learning these function classes requires involved algorithms (e.g., gradient descent with the Lasso objective), and our results show that Transformers can encode algorithms with similar performance in a single forward pass.

**Model capacity and problem dimension.** Finally, we explore how the ability of Transformers to in-context learn linear functions scales with model capacity and problem dimensionality. We find that increasing model capacity improves performance significantly, and allows the model to in-context learn higher-dimensional functions. Also, increasing the capacity often improves performance under distribution shifts significantly, even when the absolute improvement in the standard error is small.

## 2 Training models for in-context learning

We now describe a general methodology for training a model that can in-context learn a function class $\mathcal{F}$ with respect to a distribution $D_{\mathcal{F}}$ over functions, and $D_{\mathcal{X}}$ over inputs. We start by constructing random training prompts as follows. We sample a random function $f$ from the class according to $D_{\mathcal{F}}$, then create a set of random inputs $x_1, \ldots, x_{k+1}$ drawn independently from $D_{\mathcal{X}}$, and evaluate $f$ on these inputs to produce the prompt $P = (x_1, f(x_1), \ldots, x_{k+1}, f(x_{k+1}))$. For example, in the case of linear functions, inputs could be drawn from the isotropic Gaussian distribution $N(0, I_d)$, and a random function chosen by sampling weight vector $w$ from $N(0, I_d)$ and setting $f(x) = w^\top x$.

Now, given such prompts, we train a model to predict $f(x_i)$ for a given $x_i$ based on a set of in-context examples. Concretely, let $P^i$ denote the prompt prefix containing $i$ in-context examples (the first $i$ input-output pairs) and the $(i+1)^{\text{th}}$ input: $P^i = (x_1, f(x_1), x_2, f(x_2), \ldots, x_i, f(x_i), x_{i+1})$. Then, we train a model $M_\theta$ parameterized by $\theta$ to minimize the expected loss over all the prompt prefixes:

$$\min_\theta \ \mathbb{E}_P \left[ \frac{1}{k+1} \sum_{i=0}^{k} \ell \left( M_\theta \left( P^i \right), f \left( x_{i+1} \right) \right) \right], \tag{2}$$

where $\ell(\cdot, \cdot)$ is an appropriately chosen loss function. Below, we describe how this general methodology can be implemented for a concrete model family (see Appendix A for additional details).

**Model structure.** We use a decoder-only Transformer architecture [69] from the GPT-2 family [54], with 12 layers, 8 attention heads, and a 256-dimensional embedding space (22.4M parameters). The model takes as input a sequence of vectors in its embedding space and predicts the next vector in the sequence within the same space (in language modeling, these vectors correspond to input tokens). We apply this model to our prompts $(x_1, f(x_1), \ldots, x_{k+1}, f(x_{k+1}))$ as follows. We map each prompt output $f(x_i)$ to the same dimension as prompt inputs $x_i$ by appending zeros, and map the prompt inputs and outputs into the latent embedding space of the Transformer through a (learnable) linear transformation. We then use another (learnable) linear transformation to map the vector predicted by the model to a scalar. Note that the Transformer architecture allows us to compute the prediction $(M_\theta(P^i))$ for all prompt prefixes in a single forward pass.

**Training.** We train the model according to the training objective in (2) using squared error as the loss function. We sample a batch of random prompts at each training step and update the model through a gradient update. We use a batch size of 64 and train for 500k steps. This training is done from scratch, that is, we do *not* fine-tune a pre-trained language model, nor do we train on actual text.

**Curriculum learning.** Many function classes contain functions of varying complexity. We exploit this by training our model using a curriculum [5, 20, 60, 73], where we train on a simpler distribution of functions in the beginning (e.g., linear functions with weight vectors restricted to a low-dimensional subspace) and gradually increase the function complexity. This speeds up training drastically, often allowing us to train models that would be significantly more expensive to train without a curriculum (see Section 6 for details).

# 3 In-context learning of linear functions

In the previous section, we describe a general methodology for training Transformer models to in-context learn a class of functions. Here, we focus on a simple function class—namely linear functions—and study how well models trained using our methodology can in-context learn this class.

**Prompt distribution.** We consider the class of linear functions $\mathcal{F} = \left\{ f \mid f(x) = w^\top x, \; w \in \mathbb{R}^d \right\}$, in $d$ dimensions where $d = 20$. We sample $x_1, \ldots, x_k, x_{\text{query}}$, and $w$ independently from the isotropic Gaussian distribution $N(0, I_d)$. We then compute each $y_i = w^\top x_i$ and construct the prompt as $P = (x_1, \; y_1, \; x_2, \; y_2, \ldots, \; x_k, \; y_k, \; x_{\text{query}})$.

**Baselines.** To contextualize the performance of our trained model, we compare it to other learning algorithms: (a) the least squares estimator, computing the minimum-norm linear fit to the in-context examples $(x_i, \; y_i)$, (b) $n$-Nearest Neighbors, averaging the $y_i$ values for the $n$ nearest neighbors of $x_{\text{query}}$, (c) averaging the values $y_i x_i$ to estimate $w$ and compute the inner product of this estimate with $x_{\text{query}}$. Least squares is the optimal estimator for this problem and thus serves as a lower bound. The other baselines are consistent (but sub-optimal) estimators that are easier to compute and thus provide an estimate of the performance achieved by simple approaches. See Appendix A.3 for more details.

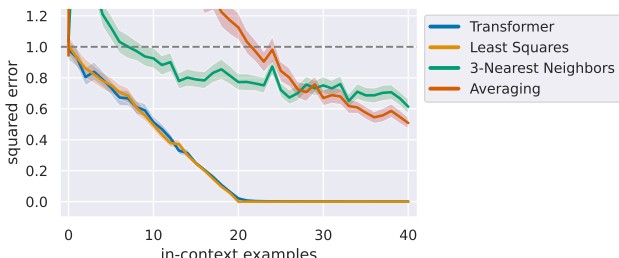

Figure 2: *Evaluating the trained Transformer on in-context learning linear functions.* We plot the normalized squared error of the Transformer $((M(P) - w^\top x_{\text{query}})^2 / d)$ and the relevant baselines, as we vary the number of in-context examples. Transformer's error decreases at a rate comparable to least squares. When the number of in-context examples reaches the problem dimension $d$ (here 20), least squares achieves 0 error while the Transformer achieves an error of 0.02, improving to 0.0006 at $2d$ examples. The simple baselines perform better than the zero estimator (dashed), but still rather poorly. (Error averaged over 1280 prompts. 90% confidence intervals, 1000 bootstrap trials.)

## 3.1 Transformers can in-context learn linear functions

We show the in-context learning ability of the resulting model along with the relevant baselines in Figure 2. The trained Transformer is able to in-context learn the class of linear functions with respect to the prompt distribution specified above, performing comparably to the optimal least squares estimator for any number of in-context examples considered. While the simpler baselines achieve non-trivial error, they are far from optimal, indicating that the trained model encodes a more complex algorithm. Moreover, in Appendix B.7, we show that the model cannot be relying on memorization of training prompts or weight vectors, and thus encodes an algorithm capable of in-context learning linear functions that are very different from those seen during training.

## 3.2 What functions is the model learning in-context?

Recall that the goal of our model is: given the prompt $P = (x_1, \; w^\top x_1, \ldots, x_k, \; w^\top x_k, \; x_{\text{query}})$, output $w^\top x_{\text{query}}$. Thus, if we fix the prefix given by the $k$ in-context examples, we can view the output of the model as a function $\hat{f}_{w, x_{1:k}}(x_{\text{query}})$, that approximates $w^\top x_{\text{query}}$. When $k < d$ (fewer in-context examples than dimensions), the ground truth cannot be recovered perfectly and the ideal model should approximate $(\text{proj}_{x_{1:k}}(w))^\top x_{\text{query}}$, where $\text{proj}_{x_{1:k}}(w)$ is the projection of $w$ onto the subspace spanned by $x_1, \ldots, x_k$. Here, we will evaluate how accurately the model approximates this.

**Visualizing along a random direction.** For a randomly sampled fixed prefix, we visualize $\hat{f}_{w, x_{1:k}}(x_{\text{query}})$ as we vary the query input along a random direction $x$ (Figure 3a). That is, we

pick a random unit vector $x$, and evaluate $\hat{f}_{w,x_{1:k}}(\lambda x)$ as we vary $\lambda$, the distance of the query input from origin. We observe that $\hat{f}_{w,x_{1:d}}(\lambda x)$ and $\hat{f}_{w,x_{1:2d}}(\lambda x)$ closely match the ground truth and $\hat{f}_{w,x_{1:d/2}}(\lambda x)$ matches the projected ground truth, when the distance from origin is not too large compared to the norm of a typical randomly sampled input. In fact, in Appendix B.1, we show that the model is quite robust to scaling the query input: the error doesn't increase much as we scale up the query input by a factor of up to 2, or scale down by a factor of up to 16, and degrades slowly after.

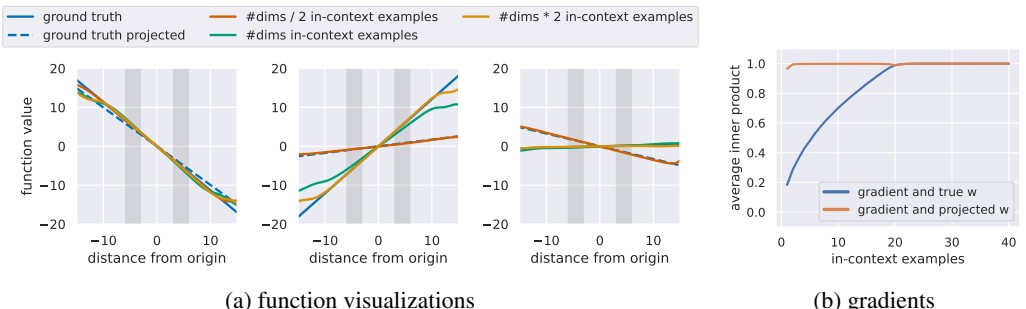

(a) function visualizations          (b) gradients

Figure 3: *Understanding the prefix-conditioned function.* (a) Model prediction as we fix the in-context examples and vary the query input along a random direction (three random prompts). The shaded regions denotes where the norm of a randomly training input lies with probability 0.99. When the scale of the query input is close to this range, the model prediction is close to the true linear function (or its projection to the space of in-context inputs when $k < d$). (b) We compute the gradient of the model prediction with respect to the query input, and plot its (normalized) inner product with the true and projected $w$, averaged over 1280 prompts. The gradient aligns almost perfectly with $w$ when $k \geq d$, and with projected $w$ for all $k$, indicating that the model locally aligns with the ground truth.

**Local correctness.** So far, we have seen that the model is able to make predictions close to the ground truth for randomly drawn query inputs and in-context examples. We will now turn our attention to the local change of $\hat{f}$ around $x_{\text{query}}$ by considering the gradient of the function $\hat{f}_{w,x_{1:k}}(x_{\text{query}})$ with respect to $x_{\text{query}}$ (our model is fully differentiable so we can compute it directly). Since $\hat{f}$ computed by the model should ideally approximate $\text{proj}_{x_{1:k}}(w)^{\top} x$, this gradient should lie in the direction of the projected ground truth $\text{proj}_{x_{1:k}}(w)$. In Figure 3b, we show the inner product between the gradient and $\text{proj}_{x_{1:k}}(w)$ (both normalized), averaged over 1280 random prompts, and observe that they align almost perfectly. Since $\text{proj}_{x_{1:k}}(w) = w$ almost surely when $k \geq d$, we observe that the gradient also aligns with $w$ perfectly in this regime. Thus the model is locally correct around the query input.

## 4 Extrapolating beyond the training distribution

In the previous section, we demonstrated that we can train a model to in-context learn linear functions with respect to the distribution of prompts encountered during training. That is, we evaluate the in-context learning ability of the model with respect to distributions $D_{\mathcal{X}}$ and $D_{\mathcal{F}}$ that were also used to train the model. Here, we evaluate the in-context learning performance of our model on prompt distributions different from the one used for training. Our overarching goal is to better understand the learning algorithm encoded by our model by analysing how it performs on such prompts.

Formally, we refer to the distribution of functions used during training as $D_{\mathcal{F}}^{\text{train}}$ and the corresponding distribution of prompt inputs as $D_{\mathcal{X}}^{\text{train}}$. During inference, functions are sampled from a (potentially different) distribution $D_{\mathcal{F}}^{\text{test}}$, while prompt inputs from a distribution $D_{\mathcal{X}}^{\text{test}}$. Moreover, deviating again from our analysis so far, we also consider a separate distribution $D_{\text{query}}^{\text{test}}$, from which the query input is sampled, potentially dependent on the rest of the in-context inputs $x_1, \ldots, x_k$ (sampled from $D_{\mathcal{X}}^{\text{test}}$).

Within this framework, we consider the same model as last section, and evaluate its performance on prompts that deviate from those encountered during training, either by

1. sampling prompt inputs or functions from a different distribution, that is $D_{\mathcal{X}/\mathcal{F}}^{\text{train}} \neq D_{\mathcal{X}/\mathcal{F}}^{\text{test}}$ or

2. introducing a mismatch between in-context examples and query input, i.e., $D_{\text{query}}^{\text{test}} \neq D_{\mathcal{X}}^{\text{test}}$.

We describe three such prompt distributions below, along with the corresponding results in Figure 4, and provide the full results in Appendix B.2. Overall, the model performs reasonably accurate in-context learning with respect to these prompt distributions, indicating that it has indeed learnt to perform linear regression to some generality.

Recall that we generate a training prompt $P = (x_1, w^T x_1, \ldots, x_k, w^T x_k, x_{\text{query}})$ by drawing the prompt inputs ($x_i$ and $x_{\text{query}}$), and the weight vector ($w$) i.i.d. from $N(0, I_d)$, with $d = 20$.

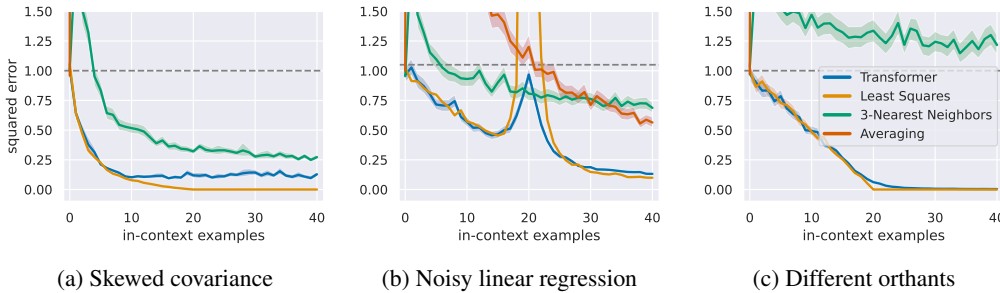

(a) Skewed covariance    (b) Noisy linear regression    (c) Different orthants

Figure 4: *In-context learning on out-of-distribution prompts.* We evaluate the model on prompts that deviate from those seen during training by: (a) sampling prompt inputs from a non-isotropic Gaussian, (b) adding label noise to in-context examples, (c) restricting in-context examples to a single (random) orthant. Model error degrades gracefully and remains close to that of the least squares estimator, indicating that its in-context learning ability extrapolates beyond the training distribution.

**Skewed covariance.** We sample prompt inputs from $N(0, \Sigma)$ where $\Sigma$ is a skewed covariance matrix with eigenbasis chosen uniformly at random and $i^{\text{th}}$ eigenvalue proportional to $1/i^2$. We normalize the inputs so that their expected squared norm is equal to that of inputs encountered during training, and study the effect of input scale separately in Appendix B.2. The model matches the performance of least squares until $k = 10$, mimicking the sharp drop in the error in this regime, but its error plateaus afterwards (see Figure 4a). Thus, it is not perfectly robust to this distribution mismatch but still does relatively well, achieving less than half the error of the nearest neighbor baseline in most cases.

**Noisy linear regression.** We add noise to each prompt output: the $i^{\text{th}}$ output is equal to $w^T x_i + \epsilon_i$ where $\epsilon_i \sim N(0, 1)$. The trained model closely tracks the performance of least squares for most prompt sizes (see Figure 4b). Interestingly, the model also exhibits the double descent error curve [2] that is known to manifest for the least squares estimator [46]. Note that in this noisy setting, the optimal estimator corresponds to least squares with appropriate $\ell_2$-regularization, which we cannot expect the model to learn since it was trained on noiseless data.

**Different orthants for in-context and query inputs.** We fix the sign of each coordinate to be positive or negative for all in-context inputs $x_i$ (at random). As a result, all in-context inputs lie in the same orthant, while the query input lies in another orthant with high probability. The model is not affected by the mismatch between in-context and query inputs and closely match the performance of least squares. In this case, the model achieves errors $0.062$ and $0.004$ for 20 and 40 in-context examples respectively (see Figure 4c), whereas recall that it achieves errors $0.02$ and $0.0006$ on standard prompts. This indicates that the model is not relying on some variant of nearest neighbor search as in that case, its error would have been higher (see the 3-nearest neighbor baseline).

## 5 More complex function classes

We now consider in-context learning for more complex function classes, namely sparse linear functions, decision trees, and two-layer ReLU neural networks. We are back in the setting where the prompt distribution during inference is same as that during training. The overall methodology remains the same: we sample random functions from these families and train a Transformer to approximate these functions given in-context examples. (See Appendix A.3 for details and baselines.) Note that, here, we are training a new Transformer model (from scratch) for each function class, independently of the model studied in the previous sections (which was trained to in-context learn linear functions).

**Sparse linear functions.** First, we consider functions of the form $f(x) = w^\top x$ where $w \in \mathbb{R}^d$ and has exactly $s$ non-zero coordinates. To sample a prompt $P = (x_1, f(x_1), \ldots, x_k, f(x_k), x_{\text{query}})$, we

draw prompt inputs $x_i$ and $x_{\text{query}}$, and a weight vector $w$ from $N(0, I_d)$, and then zero out all but $s$ coordinates of $w$ uniformly at random. In this setting, the least squares estimator is no longer optimal. One can perform better by leveraging sparsity, e.g., using Lasso [68], which involves solving the least squares objective with an $\ell_1$-norm weight regularizer. We plot the performance of our model trained for $d = 20$ and $s = 3$ in Figure 5a, and observe that it is also able to leverage sparsity, nearly matching the performance of Lasso. Note that, unlike least squares, Lasso does not have a closed form expression and involves iterative minimization of the regularized objective, yet the Transformer is able to achieve comparable performance in a single forward pass.

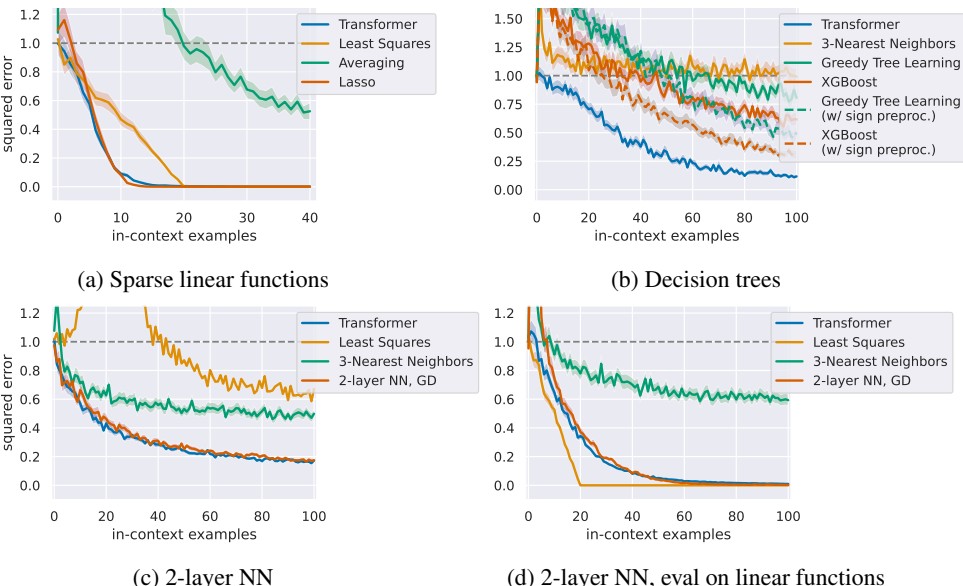

(a) Sparse linear functions          (b) Decision trees

(c) 2-layer NN          (d) 2-layer NN, eval on linear functions

Figure 5: *Training a Transformer to in-context learn more complex function classes.* (a) A Transformer trained on prompts generated using sparse linear functions can in-context learn this class, with error decreasing at a rate similar to Lasso, and significantly better than minimum norm least squares. (b) A Transformer trained on prompts generated using random decision trees can in-context learn this class, with much better performance than greedy tree learning or tree boosting. (c) A Transformer trained on prompts generated using random 2-layer ReLU neural networks can in-context learn this class. The error decreases at a rate similar to the baseline which involves training a neural network using a variant of gradient descent with in-context examples as the training data. (d) The same model (from (c)) can in-context learn the class of linear functions. The error decreases at a rate slower than least squares, but comparable to a neural network trained using a variant of gradient descent. All errors are normalized so that the zero estimator achieves an error of 1 (dashed).

**Decision trees.** Next, we consider the class of depth 4 decision trees with 20 dimensional inputs. A function $f$ in this class is represented by a full binary tree (16 leaves) where each non-leaf node is associated with a coordinate, and each leaf is associated with a value. To evaluate $f$ on an input $x$, we traverse the tree starting from the root node, going to the right if the coordinate associated with that node is positive and to the left otherwise. $f(x)$ is given by the value associated with the leaf node reached. To sample a random prompt $P = (x_1, f(x_1), \ldots, x_k, f(x_k), x_{\text{query}})$, we draw inputs $x_i$s and $x_{\text{query}}$ from $N(0, I_d)$, and $f$ corresponds to a tree where the coordinates associated with non-leaf nodes are drawn uniformly at random from $\{1, \ldots, d\}$ and the values associated with the leaves are drawn from $N(0, 1)$. In Figure 5b, we show that Transformers can be trained to in-context learn this class much better than greedy tree learning and boosting (via XGBoost [13]). Since the decision trees in our class of functions predict solely based on the sign pattern of $x_i$s, we also consider a baseline where we provide the greedy learning and XGBoost algorithms with the signs of each $x_i$ instead. This significantly improves their performance, but they still perform much worse than the Transformer (error 0.31 vs. 0.12 at 100 in-context examples).

**Two-layer neural networks.** Finally, we consider the class of two layer ReLU networks containing functions of the form $f(x) = \sum_{i=1}^{r} \alpha_i \ \sigma(w_i^\top x)$, where $\alpha_i \in \mathbb{R}$, $w_i \in \mathbb{R}^d$ and $\sigma(\cdot) = \max(0, \cdot)$ is the ReLU activation function. To draw a random prompt $P =$

$(x_1, f(x_1), \ldots, x_k, f(x_k), x_{\text{query}})$, we sample inputs $x_i$s and $x_{\text{query}}$ from $N(0, I_d)$, along with network parameters $a_i$s and $w_i$s from $N(0, 2/r)$ and $N(0, I_d)$ respectively. We set the input dimension $d$ to 20 and the number of the hidden nodes $r$ to 100. In Figure 5c, we show that Transformers can be trained to in-context learn this class of functions. In fact, the Transformer performs comparably to the baseline which trains a two-layer neural network of the same architecture on in-context examples using Adam [31], a variant of gradient descent (see Appendix A.3 for details). Moreover, the model trained to in-context learn two-layer neural networks is also able to in-context learn linear functions (for which it is not explicitly trained), albeit at a rate slower than least squares, but comparable to a neural network trained on in-context examples generated using a linear function. (see Figure 5d).

## 6    Investigating what matters for in-context learning

We now return to the setting of training models to in-context learn linear functions and explore different factors that lead to successful in-context learning.

**Problem Dimension and Capacity.** In Section 3 and 4, we saw that Transformer models can be trained to in-context learn 20-dimensional linear functions accurately and relatively robustly. To explore the interplay between problem dimensionality and capacity, we also consider models with fewer parameters (see Appendix A.1) and train each architecture on $\{10, 30, 40, 50\}$-dimensional problems. In Figure 6, we plot the model error with $2d$ in-context examples as we vary the problem dimension $d$ and the model capacity. In the standard setting, i.e., when the training and inference time prompt distributions are the same, we observe that the error decreases as we increase the capacity or reduce the problem dimensionality (see Figure 6a)—i.e., model capacity helps. For out-of-distribution prompts, the settings with skewed covariance or with in-context example and query inputs lying in different orthants are challenging, especially for higher dimensional problems. However, the error decreases considerably (in most cases) as we increase the model capacity, even when absolute decrease in the standard error is small (see Figure 6b and 6c). See Appendix B.3 for additional plots.

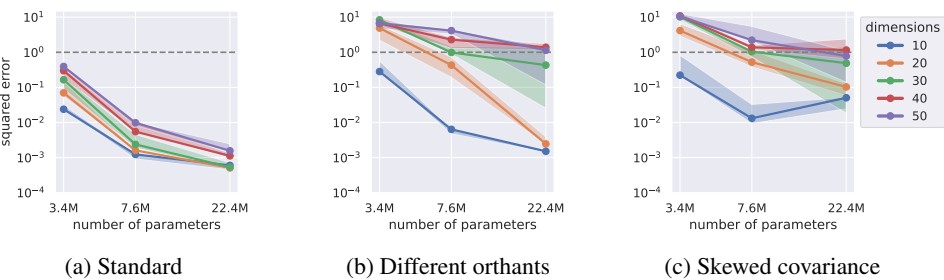

(a) Standard          (b) Different orthants          (c) Skewed covariance

Figure 6: *Understanding the effect of model capacity and problem dimension on in-context learning performance for in-distribution (a) and out-of-distribution (b,c) prompts.* We train Transformers to in-context learn linear functions and plot the error with $2d$ in-context examples as we vary problem dimension $d$ and model capacity. Capacity helps in most cases, especially on out-of-distribution prompts, even when the absolute gains in the in-distribution setting are small. We train 3 models in each case with different random seeds, and show the median error (solid lines), and the minimum and maximum errors (shaded region). (See Appendix B.4 for training variance analysis.)

**Curriculum learning.** When training our models, we initially draw the prompt inputs from a fixed 5 dimensional subspace (by setting some of the coordinates to 0) with prompt length 11 (number of input-output pairs), and increase the subspace dimension by 1 and prompt length by 2 every $2,000$ training steps, until the subspace dimension reaches the ambient dimension $d$ and prompt length reaches $2d + 1$ (see Appendix A.2 for details). This speeds up training drastically, especially for higher dimensional problems: for dimension 50, the loss barely decreases through the 500k training steps without curriculum but reaches close to the optimum with curriculum. For the 20 dimensional problem where we were able to train the model without curriculum within the training (step count) budget, we did not observe any qualitative difference in accuracy or robustness compared to the model trained with curriculum. We compare training with and without curriculum in Appendix B.5.

Notably, when training Transformers without curriculum, there is an initial—relatively long—period in training where the loss does not decrease, followed by a period of sharp decrease. The length of

this period varies with training randomness and seems to increase on average with problem dimension. Interestingly, Olsson et al. [48] observe a similar jump in the in-context learning ability of a language model which they attribute to the formation of "induction heads".

**Number of distinct prompts or functions seen during training.** To estimate the amount of training data required for in-context learning, we perform two ablation studies. In the first study, we limit the number of distinct prompts seen during training. That is, we create a set of $n_p$ randomly generated prompts (as described in Section 2), and sample prompts from this set during training (here, we train without curriculum, as it would introduce additional prompts during the warmup phase). In the second study, we only limit the number of distinct functions used for training. That is we create a set of $n_w$ randomly chosen vectors (corresponding to $n_w$ linear functions) and sample weight vectors uniformly from that set to generate the training prompts (the inputs are still sampled from $N(0, I_d)$ for each training prompt). We find that the amount of training data required is relatively small: non-trivial in-context learning is possible with $n_p = 100\text{k}$ or $n_w = 1\text{k}$, and the error drops close to that of the unrestricted model (discussed in Section 3) with $n_p = 1\text{M}$ or $n_w = 10\text{k}$ (details in Appendix B.6). For context, in Section 3, the model is trained on fresh prompts at each step and thus encounters 32M distinct linear functions and prompts (500k training steps with a batch size of 64).

# 7   Related work

**In-context learning.** Since Brown et al. [10] demonstrated the in-context learning ability of GPT-3, there has been a significant interest in improving and understanding this capability [36, 39, 79, 38, 59, 40, 14, 43, 33]. The works most relevant to ours are as follows. Xie et al. [74] propose a Bayesian inference framework explaining how in-context learning works despite formatting differences between training and inference distributions. Razeghi et al. [57] show that in-context learning for numerical reasoning tasks is better for instances whose terms are more prevalent in training data. Min et al. [39] demonstrate tasks where in-context learning works even when the prompt outputs are chosen randomly, questioning to what extent these models are truly learning new tasks on-the-fly, while Rong [58] gives examples of novel tasks on which these models demonstrate on-the-fly learning ability. Chan et al. [12] demonstrate that distributional properties such as long-tailedness are crucial for in-context learning on an image-based few-shot dataset. Olsson et al. [48] and Elhage et al. [19] consider a different framing of in-context learning, referring to any model behavior that utilizes information in a prompt to make predictions that improve with prompt size. They hypothesize the existence of special circuits inside Transformer models responsible for in-context learning, that copy similar patterns from the prompt sequence. Pesut [52] and Dinh et al. [16, Table 16] consider in-context learning for small tabular datasets and learning problems in one and two dimensions, and show that GPT-3 can obtain non-trivial accuracy. Our work contributes to this line of work, by posing in-context learning as a well-defined problem of learning function classes at inference time, and empirically investigating if we can train models that in-context learn simple function classes.

**Transformers.** There is a long line of work investigating the capabilities [69, 15, 77, 51, 76, 7, 78], limitations [25, 6], applications [37, 17, 49], and internal workings [19, 65, 71, 18, 48] of Transformer models. Most similar to our work, Müller et al. [44] and Nguyen and Grover [47] demonstrate the ability of Transformer models to solve prediction tasks using the input context, albeit in different settings. Müller et al. [44] introduce a "Prior-data fitted transformer network" that is trained to approximate Bayesian inference with priors such as Gaussian processes and Bayesian neural networks, and use it to perform downstream tasks such as tabular dataset classification and few-shot image classification. Nguyen and Grover [47] introduce Transformer neural processes, building on prior work on neural processes [24, 23, 30], and show that they achieve state-of-the art performance on tasks such as image completion and contextual multi-armed bandits. Our work complements these works, focusing on understanding the in-context learning ability of Transformers for various simple function classes and the extent to which this ability extrapolates beyond the training distribution.

**Meta-learning.** Training a model to perform in-context learning can be viewed as an instance of the more general learning-to-learn or meta-learning paradigm [62, 45, 67]. Typical approaches from this extensive line of work (see [28] for a survey) include: training a meta-learner to update the parameters of a downstream learner [4, 34], learning parameter initializations which allow to quickly train for downstream tasks [21, 56], learning latent embeddings for effective similarity search [66]. Most relevant to our setting are approaches that directly take as input examples from a downstream

task and a query input and produce the corresponding output [26, 42, 61, 24, 23, 32]. Our work contributes to this line of work, by investigating the learning-to-learn abilities of Transformer models.

**Data-driven algorithm design.** Another line of work aims to discover algorithms that perform well on a distribution of inputs [27, 75, 70, 3, 29, 64, 63] (as opposed to algorithms with guarantees on their worst-case performance). See Balcan [1] for a survey on advancements on the theoretical foundations of such algorithms. Our work can be viewed as part of this line of work, as we train Transformer models to discover algorithms for different learning problems.

## 8 Discussion

In this work, we formalize and study the question: can we train models that learn different classes of functions in-context? We show that Transformer models trained from scratch can in-context learn the class of linear functions, with performance comparable to the optimal least squares estimator, even under distribution shifts. Moreover, we show that in-context learning is also possible for sparse linear functions, decision trees, and two-layer neural networks; learning problems which are solved in practice with involved iterative algorithms such as gradient descent.

At the same time, understanding the implications of our results for language models requires further investigation. A pertinent question regarding the in-context learning capabilities of language models is how they leverage in-context examples [41]. Our results demonstrate that Transformers can encode complex learning algorithms that utilize in-context examples in a far-from-trivial manner. In fact, this is the case for standard Transformer architectures trained with standard optimization procedures. The extent to which such non-trivial in-context learning behavior exists in large language models is still open, but we believe that our work takes a step towards formalizing and understanding this question.

Our work lays the groundwork for several future directions.

**Complexity of in-context learning.** We empirically show that model capacity helps in performing in-context learning accurately and robustly. How does the in-context learning loss (1) depend on the complexity of the function class $\mathcal{F}$, the capacity of model $M$, and data used to train $M$. Understanding this question for models explicitly trained to perform in-context learning may suggest an upper bound for the in-context learning performance of models such as GPT-3 that are not explicitly trained for it.

**Curriculum learning.** Within our framework, there is natural notion of curriculum learning where, during training, we gradually increase the complexity of the function class learned in-context. This leads to drastic training speed-ups. What is the reason behind such a speedup? Are similar speedups also possible for training large language models (thus significantly reducing training time and energy)?

**Inductive bias of model families.** Our framework presents an opportunity to understand and compare the inductive biases of different model families (e.g., Transformers vs. LSTMs) in a well-defined setting. For instance, a concrete question is: Are there function classes that are easier to in-context learn using Transformers but harder for LSTMs and vice-versa?

**Understanding the learning algorithms encoded in Transformers.** The models we train can perform in-context learning, and are thus themselves encoding learning algorithms. However, we do not really understand the encoded algorithms. It would thus be worth investigating the internal workings of these models to get a better understanding of these algorithms. Moreover, for settings such as decision trees, we do not have a good understanding of what the optimal learning algorithms are or when known heuristics work [9, 11]. Nevertheless, in Section 5 we found that Transformers are able to discover sample efficient algorithms, thus suggesting an intriguing possibility where we might be able to discover better learning algorithms by reverse engineering these models.

## Acknowledgements

We thank Niladri Chatterji, Micah Goldblum, Rohith Kuditipudi, Shibani Santurkar, Carmen Strassle, Mirac Sugzun, Li-Yang Tan, and anonymous reviewers for helpful comments.

SG was funded by a Stanford Interdisciplinary Graduate Fellowship. DT was funded by Open Philanthropy, and partially supported by NSF Award CCF-1813049. GV was supported by NSF Awards CCF-1704417, CCF-1813049, Frontier Award 1804222 and DOE award DE-SC0019205. We performed our experiments on the Stanford NLP cluster.

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
