# A Experimental setup

Here, we provide additional details on our experimental setup.

## A.1 Model architecture

We use architectures from the GPT-2 family [53] as implemented by HuggingFace [72] [3] . Specifically, we consider the following set of configurations.

| Model | Embedding size | #Layers | #Heads | (Total parameters) |
|---|---|---|---|---|
| Tiny | 64 | 3 | 2 | 3.4M |
| Small | 128 | 6 | 4 | 7.6M |
| **Standard** | 256 | 12 | 8 | 22.4M |

We use the Standard model for the bulk of our experiments and only consider the smaller models for the capacity explorations in Section 6 and Appendix B.3. Since we train on each input once (we sample new inputs at each training step), overfitting to the training data is not an issue. Therefore, we set the Dropout probability to 0.

Out of the box, these models take as input a sequence of vectors in embedding space and output a sequence of vectors in the same space. However, the tasks we study are functions from a lower dimensional vector space (e.g., 10-50 dimensions) to a scalar value. Thus, in order to use a prompt such as $x_1, f(x_1), x_2, f(x_2)\ldots$, we need to map $x_i$s and $f(x_i)$s to vectors in embedding space. We do so by first turning the scalars $f(x_i)$ into vectors of the same dimension as $x_i$ by appending 0s and then applying a learnable linear transformation to map all these vectors into the embedding space. Finally, we map the model output into a scalar value through a dot product with a learnable vector.

We treat the prediction of the model at the position corresponding to $x_i$ (that is absolute position $2i - 1$) as the prediction of $f(x_i)$. Due to the structure of these models, this prediction only depends on $(x_j, f(x_j))$ for $j < i$ and $x_i$. We ignore the model predictions at positions corresponding to $f(x_i)$.

## A.2 Training

Each training prompt is produced by sampling a random function $f$ from the function class we are training on, then sampling inputs $x_i$ from the isotropic Gaussian distribution $N(0, I_d)$ and constructing a prompt as $(x_1, f(x_1), \ldots, x_k, f(x_k))$. Given a prompt, we obtain model predictions $\hat{y}_i$ (meant to approximate $f(x_i)$) for each input, and compute the loss

$$\frac{1}{k} \sum_{i=1}^{k} (\hat{y}_i - f(x_i))^2 .$$

At each training step, we average the loss over a batch of randomly generated prompts (with different functions and prompt inputs), and perform an update step. We use the Adam optimizer [31], and train for 500,000 total steps with a batch size of 64. We use a learning rate of $10^{-4}$ for all function classes and models.

**Curriculum learning.** To accelerate training, we start by training on prompt inputs $x_i$ lying in a smaller dimensional subspace, and with fewer inputs per prompt, and gradually increase the subspace dimension and number of prompt inputs. Specifically, we zero out all but the first $d_{\text{cur}}$ coordinates of $x_i$, sample prompts of size $k_{\text{cur}}$ and leave the rest of the training process the same. We use the same schedule for all training runs for the function classes of linear functions and sparse linear functions, starting with $d_{\text{cur}} = 5$, $k_{\text{cur}} = 11$, and increasing $d_{\text{cur}}$ and $k_{\text{cur}}$ by 1 and 2 respectively, every 2000 steps, until $d_{\text{cur}} = d$, $k_{\text{cur}} = 2d + 1$. We use a slightly different schedule for 2 layer neural networks and decision trees as we want prompts with more inputs for these function classes. For these classes, we start with $d_{\text{cur}} = 5$, $k_{\text{cur}} = 26$, and increase $d_{\text{cur}}$ and $k_{\text{cur}}$ by 1 and 5 respectively, every 2000 steps, until $d_{\text{cur}} = d$, $k_{\text{cur}} = 5d + 1$.

Overall, with curriculum, a training prompt $(x_1, f(x_1), \ldots, x_{k_{\text{cur}}}, f(x_{k_{\text{cur}}}))$ is generated by sampling a random function $f$ from the function class, drawing inputs $x_i$ by sampling i.i.d. from $N(0, I_d)$ and

---

[3] https://huggingface.co/docs/transformers/model_doc/gpt2

zeroing out all but the first $d_{\text{cur}}$ coordinates. Given model predictions $\hat{y}_i$, the loss is given by

$$\frac{1}{k_{\text{cur}}} \sum_{i=1}^{k_{\text{cur}}} (\hat{y}_i - f(x_i))^2 .$$

**Sampling random functions.** For the class of linear functions, we sample random function $f(x) = w^\top x$ by drawing $w \sim N(0, I_d)$. For our main setting (Section 3 and 4), we set $d = 20$.

For the class of two-layer neural networks, we sample $f(x) = \sum_{i=1}^{r} \alpha_i \sigma(w_i^\top x)$, where $\alpha_i$s and $w_i$s are drawn i.i.d. from $N(0, 2/r)$ and $N(0, I_d)$ respectively. We set $d = 20$ and $r = 100$.

For the class of $k$-sparse linear functions, we sample $f(x) = w^\top x$ by drawing $w \sim N(0, I_d)$ and zeroing out all but $k$ coordinates of $w$ chosen uniformly at random from the first $d_{\text{cur}}$ coordinates (as defined in the curriculum learning description above). We set $d = 20$ and $k = 3$.

For the class of decision trees, the random function $f$ is represented by a decision tree of depth $4$ (with 16 leaf nodes), with 20 dimensional inputs. Each non-leaf node of the tree is associated with a coordinate selected uniformly at random from $\{1, 2, \ldots, d\}$, and each leaf node is associated with a value drawn randomly from $N(0, 1)$. To evaluate $f$ on an input $x$, we traverse the tree starting from the root node, and go to the right child if the coordinate associated with the current node is positive and go to the left child otherwise. $f(x)$ is given by the value associated with the leaf node reached at the end.

**Computational resources.** We train using a single NVIDIA GeForce RTX 3090 GPU and most training runs take 5-20 hours depending on model size and context length. For instance, for the class of linear functions, training the standard model takes 17 hours for $d = 50$, 7 hours for $d = 20$ and 5.5 hours for $d = 10$. For decision trees, training the standard model takes 17 hours. The time it takes for decision trees and 50 dimensional linear functions is higher due to larger context lengths (we train for $d$ dimensional linear functions with $2d + 1$ input-output pairs per prompt).

### A.3    Baselines

**Least squares.** Minimum norm least squares is the optimal estimator for the linear regression problem. Given a prompt $P = (x_1, y_1, \ldots, x_k, y_k, x_{\text{query}})$, let $X$ be a $k \times d$ matrix with row $i$ given by $x_i$, and let $y$ be a $k$ dimensional vector with the $i^{\text{th}}$ entry $y_i$. Set $\hat{w}^T = X^+ y$, where $X^+$ denotes the Moore-Penrose pseudoinverse of $X$. The estimator predicts $M(P) = \hat{w}^T x_{\text{query}}$.

**Averaging estimator.** This corresponds to $M(P) = \hat{w}^T x_{\text{query}}$ where $\hat{w} = \frac{1}{k} \sum_{i=1}^{k} x_i y_i$. This estimator is consistent (yet sub-optimal) when $x_i$s are drawn from $N(0, I_d)$. Unlike least squares, this estimator does not involve an inverse computation, and might be easier for a model to encode.

**Nearest neighbors.** This corresponds to setting $M(P) = \frac{1}{n} \sum_{i \in S} y_i$. Here, $S$ is the set of indices of the $n$ nearest neighbors of $x_{\text{query}}$ among $x_1$ to $x_k$. For $k < n$, we average over all the $y_i$s from 1 to $k$, and for $k = 0$, we set $M(P) = 0$. We consider the nearest neighbors baselines as it might be easier for a Transformer model to encode using self-attention compared to least squares.

**Lasso.** We use this baseline for sparse linear functions (Section 5). This corresponds to $M(P) = \hat{w}^T x_{\text{query}}$, where $\hat{w}$ minimizes the $\ell_1$-norm regularized least squares objective:

$$\min_{\hat{w}} \frac{1}{2k} ||y - X\hat{w}||_2^2 + \alpha ||\hat{w}||_1.$$

We try different values of $\alpha \in \{1, 10^{-1}, 10^{-2}, 10^{-3}, 10^{-4}\}$, and report the best solution (achieving the smallest error with 10 in-context examples) corresponding to $\alpha = 10^{-2}$. To solve the optimization problem, we use the Lasso implementation from Scikit-learn [50] [4].

**Greedy Tree Learning.** We use this baseline for the class of decision trees. This corresponds to greedily learning a decision tree using the in-context examples, and using it to classify the query input. To construct the tree, at each node (starting from a root node), we choose a coordinate for partitioning the examples into two sets, so as to minimize the variance of $y_i$s in each set, averaged across the two sets. The value associated with a leaf node is the average $y_i$ value of the examples belonging to it. We use Scikit-learn's decision tree regressor [50] [5] implementation for this, with all the arguments

---

[4] https://scikit-learn.org/stable/modules/generated/sklearn.linear_model.Lasso.html

[5] https://scikit-learn.org/stable/modules/tree.html#regression

set to their default value except the max_depth argument which is set to 2. We considered values $\{1, 2, 3, 4, 5, 6, \text{unbounded}\}$ for the maximum depth and chose the value that performs best at 100 in-context examples which was 2 (which differs from the decision trees sampled from the function class which have depth 4). We also considered a baseline where we learn this tree using only the signs of each $x_i$ coordinate—after all, the decision tree we are trying to learn depends only on the signs of $x_i$. In this case, we found the optimal depth to be 4.

**Tree boosting.** For the class of decision trees, we also consider a tree boosting baseline that corresponds to learning an ensemble of decision trees (see Friedman [22] for a description of the general framework). Specifically, we use the XGBoost library [13] [6], an implementation commonly used for a wide range of real-world machine learning tasks. We performed a hyperpameter search by considering $\{1, 2, 5, 10, 50, 100, 200, 400\}$ estimators in the ensemble (equivalent to number of boosting rounds), a learning rate of $\{0.001, 0.01, 0.1, 0.3, 0.6, 1, 3\}$, and a maximum depth of $\{1, 2, 3, 4, 6, 10, 16\}$. In general, we found the performance of the learning algorithm to be quite robust. We chose the hyperparameters obtaining the best performance with 100 training examples, corresponding to 50 estimators, a maximum depth of 4, and a learning rate f 0.1. We found these hyperparameters to also be optimal when learning based on the signs of each $x_i$.

**Learning neural networks with gradient descent.** We use this baseline for the class of two-layer neural networks (Section 5). This corresponds to training a two-layer neural network on the in-context examples, and outputting its prediction on the query point. That is, $M(P) = \hat{f}(x_{\text{query}})$, where

$$\hat{f}(x_{\text{query}}) = \sum_{i=1}^{r} \hat{\alpha}_i \, \sigma(\hat{w}_i^\top x_{\text{query}}).$$

Here, $\sigma(\cdot)$ is the ReLU activation. We find parameters $\hat{\alpha}_i, \hat{w}_i$ by minimizing the squared error of the prediction for the in-context examples

$$\sum_{i=1}^{k} \left( \hat{f}(x_i) - f(x_i) \right)^2,$$

using the Adam optimizer. We use a batch size of 10 (we use full batch when the number of in-context examples is less than 10) with 5000 optimization steps, and set $r = 100$. We use a learning rate of $5 \cdot 10^{-3}$ in the case when the data is generated using a neural network, and a learning rate of $5 \cdot 10^{-2}$ when the data is generated using a linear function. We consider the setting with 100 in-context examples, and do a hyperparameter grid search over learning rate $\in \{5 \cdot 10^{-4}, 5 \cdot 10^{-3}, 5 \cdot 10^{-2}, 5 \cdot 10^{-1}, 5\}$, $r \in \{100, 400\}$, batch size $\in \{10, 100\}$, optimization algorithm $\in \{\text{adam}, \text{sgd}\}$. All the hyperparameter settings in this grid led to a similar or worse performance compared to the hyperparameter setting we choose.

---

[6] https://github.com/dmlc/xgboost

# B  Additional experimental results

## B.1  Robustness to query scale

In Figure 7, we show that the trained model is quite robust to scaling the query input (while keeping the in-context examples fixed): the error does not increase much as we scale up the query input by a factor of up to 2, or scale down by a factor of up to 16, and degrades slowly after that.

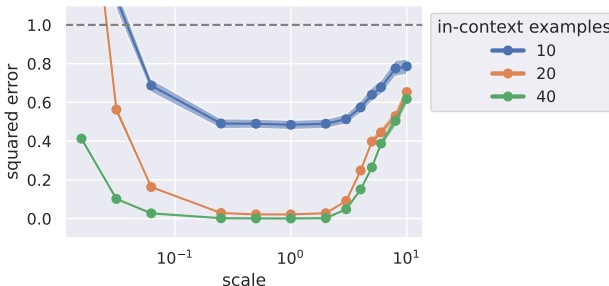

Figure 7: *Robustness to the scale of query input.* For a fixed set of in-context examples, we measure the model's error as we scale the query input by a scalar.

## B.2 Out-of-distribution prompts

Here, we describe the structure of our out-of-distribution prompts (cf. Section 4), and show the corresponding plots (Figure 8). To avoid conflating factors, we normalize the prompt inputs such that their expected norm is equal to the expected norm of inputs during training and investigate the role of scaling these inputs separately. We summarize how these prompts deviate from those seen during training in the table below.

| Prompting strategy | $D_{\mathcal{X}}^{\text{train}} \neq D_{\mathcal{X}}^{\text{test}}$ | $D_{\mathcal{F}}^{\text{train}} \neq D_{\mathcal{F}}^{\text{test}}$ | $D_{\text{query}}^{\text{test}} \neq D_{\mathcal{X}}^{\text{test}}$ |
|---|---|---|---|
| Skewed covariance | ✓ | | |
| $d/2$-dimensional subspace | ✓ | | |
| Scale inputs | ✓ | | |
| Noisy output | | ✓ | |
| Scale weights | | ✓ | |
| Different Orthants | ✓ | | ✓ |
| Orthogonal query | | | ✓ |
| Query matches example | | | ✓ |

**Skewed covariance.** (Figure 8a) We sample prompt inputs from $N(0, \Sigma)$ where $\Sigma$ is a skewed covariance matrix with eigenbasis chosen uniformly at random and $i^{\text{th}}$ eigenvalue proportional to $1/i^2$. The model matches the performance of least squares until $k = 10$, mimicking the sharp drop in the error in this regime, but its error plateaus afterwards (see Figure 4a). Thus, it is not perfectly robust to this distribution mismatch but still does relatively well, achieving less than half the error of the nearest neighbor baseline in most cases.

**Low-dimensional subspace.** (Figure 8b) We sample prompt inputs from a random $d/2$ dimensional subspace ($d = 20$ in Figure 8b). That is, we pick a random $d/2$ dimensional subspace, and draw the prompt inputs from an isotropic Gaussian distribution restricted to this subspace. In this case, the model achieves low error after 10 in-context examples, closely matching the behavior of the optimal least squares estimator (the model achieves an error of 0.036, 0.0014, and 0.00057 at 10, 20, and 40 in-context examples respectively). Crucially, unlike the training prompts, when $k$ is between 10 and 20, the prompt inputs are linearly dependent, and a model achieving low error in this regime indicates that it encodes a valid orthogonalization procedure for these inputs.

**Prompt scale.** (Figure 9) We consider the setting where the prompt scale between training and inference is different. We either scale the prompt inputs or the weight vectors, by a factor $\{1/3, 1/2, 2, 3\}$. The model is relatively robust when scaling the weight vector, but not as robust when scaling the prompt inputs, especially for the more extreme scales $1/3$ and 3. Specifically, for 40 in-context examples, the model achieves errors $0.0012, 0.0008, 0.0016, 0.0278$ when scaling the weights, and errors $0.30, 0.013, 0.043, 0.58$ while scaling the inputs, by factors $1/3, 1/2, 2$ and 3 respectively. For context, recall that with 40 in-context examples, the least squares estimator achieves an error of 0 whereas the model achieves an error of 0.0006 at the original scale.

**Noisy linear regression.** (Figure 8c) We add noise to each prompt output, that is, the $i^{\text{th}}$ output is equal to $w^T x_i + \epsilon_i$ where $\epsilon_i \sim N(0, d/20)$. The trained model closely tracks the performance of least squares for most prompt sizes. Interestingly, the model also exhibits the double descent error curve [2] that is known to manifest for the least squares estimator [46]. Note that in this noisy setting, the optimal estimator corresponds to least squares with appropriate $\ell_2$-regularization, which we cannot expect the model to learn from noiseless data.

**Different orthants for in-context and query inputs.** (Figure 8f) We fix the sign of each coordinate to be positive or negative for all in-context inputs $x_i$ (at random), and draw $x_{\text{query}}$ (as before) i.i.d. from $N(0, I_d)$. As a result, all in-context inputs lie in the same orthant, while the query input lies in another orthant with high probability. The model is not affected by the mismatch between in-context and query inputs and closely match the performance of least squares. In this case, the model achieves errors 0.062 and 0.004 for 20 and 40 in-context examples respectively, whereas recall that it achieves errors 0.02 and 0.0006 on standard prompts. This indicates that the models is not relying on some variant of nearest neighbor search as in that case, its error would have been higher (see the 3-nearest neighbor baseline).

**Query input orthogonal to in-context inputs.**(Figure 8d) We choose the query input randomly in the space orthogonal to the space spanned by in-context example inputs (there can be at most $d-1$ in-context examples for an orthogonal query to exist). That is, we draw the query input from an isotropic Gaussian distribution restricted to the subspace orthogonal to the space spanned by the in-context examples. Here, there is no information relevant to the query input in the in-context examples and thus the model would ideally predict something close to 0 to minimize the error. Indeed, the model outputs such a prediction, achieving an error close to 1.

**Query input matches an in-context example.**(Figure 8e) We set the query input equal to one of the in-context examples chosen uniformly at random. Thus it's possible to achieve zero error since the in-context examples include the correct prediction for the query input already. In this case, the model achieves errors $0.001, 0.001, 0.0005$ for $10, 20, 40$ examples respectively thus making close to the correct prediction, without being affected by the additional in-context examples present.

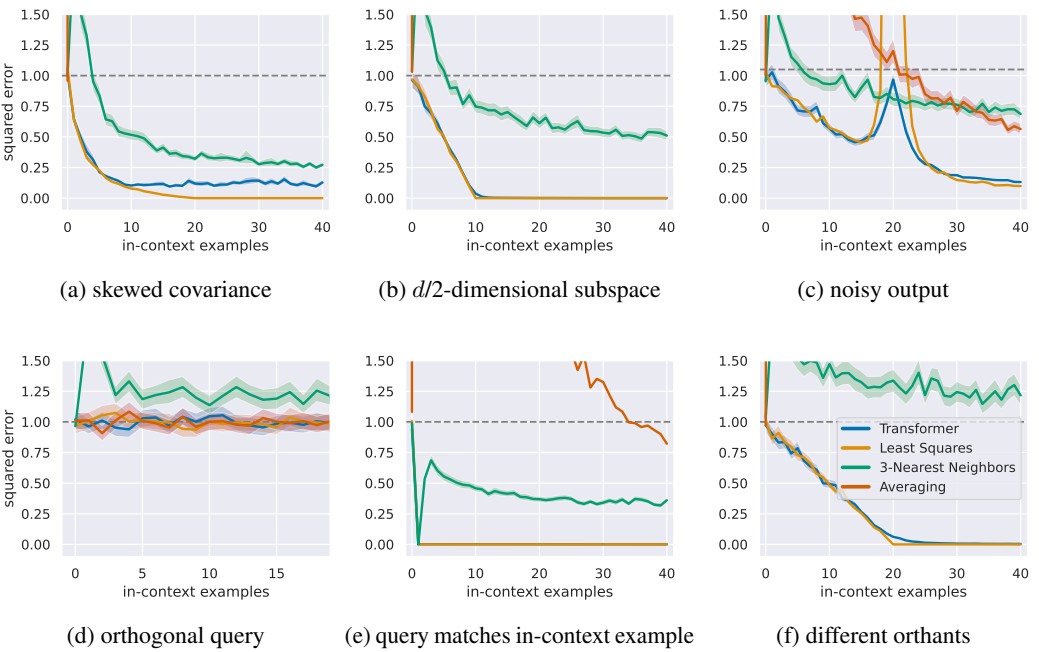

Figure 8: *In-context learning on out-of-distribution prompts.* We evaluate the model trained to in-context learn linear functions on prompt distribution that deviates from the training prompt distribution. In general, the model error degrades gracefully and closely tracks that of the least squares estimator.

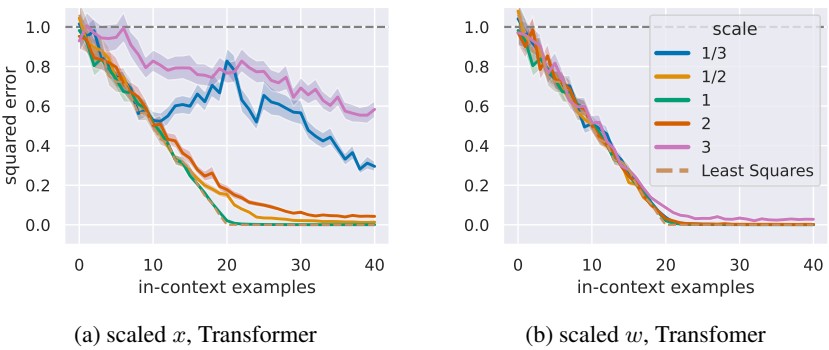

Figure 9: *In-context learning robustness to prompt scaling.* We evaluate the model trained to in-context learn linear functions when we scaled the prompt inputs $x$ or the weight of the function class $w$. The model appear to be quite robust to scaling $w$ but their performance degrades when scaling the inputs up or down by a factor of 3.

## B.3 Effect of problem dimension and model capacity

We plot the model error for additional out-of-distribution prompts in Figure 10 for $2d$ in-context examples (with the exception of orthogonal queries where we use $d-1$ in-context examples).

Similar to the settings in Section 6 (skewed covariance and different orthants), accuracy improves with capacity in most cases. One exception is scaling $x$ (Figure 10e), in which case we do not see any clear trend. In the case of noisy output (Figure 10b), the accuracy almost saturates at 7.6M parameters, close to the error of the least squares estimator. In the case of orthogonal query input (Figure 10c), the model achieves the optimum error of 1 even with the tiny model with 3.4M parameters.

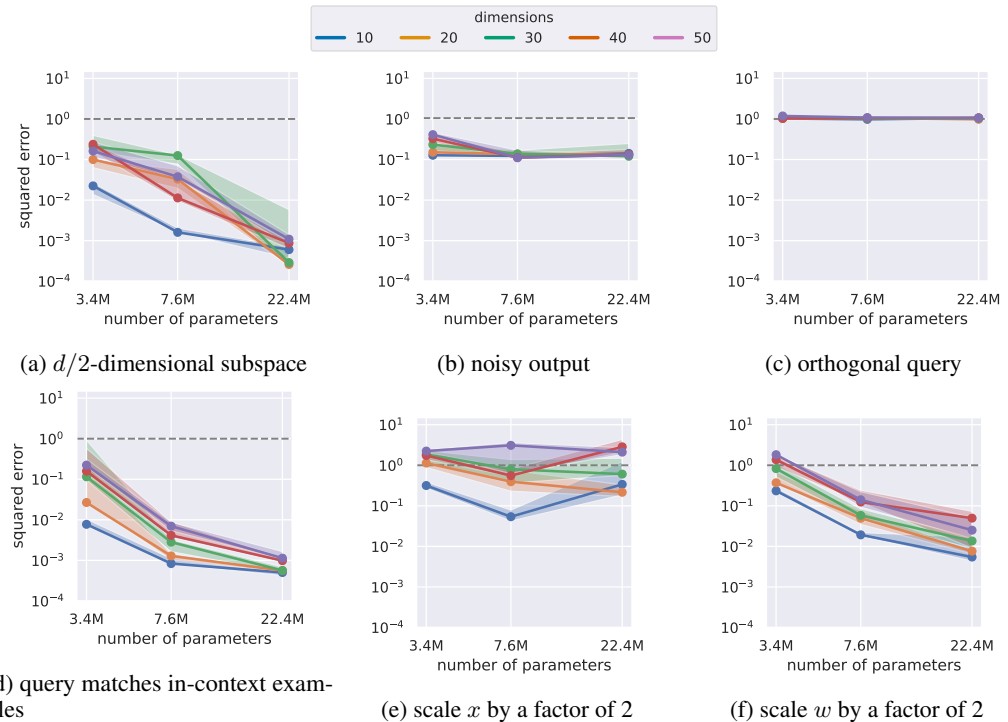

(a) $d/2$-dimensional subspace

(b) noisy output

(c) orthogonal query

(d) query matches in-context examples

(e) scale $x$ by a factor of 2

(f) scale $w$ by a factor of 2

Figure 10: *The effect of model capacity and problem dimension for in-context learning performance on out-of-distribution prompts.* We train Transformer models of varying capacity to in-context learn linear function in varying dimensions $d$. We plot the error with $2d$ in-context examples (or $d-1$ for orthogonal queries). We find that capacity helps in most cases, with the exception of scaling $x$ where we find no clear trend. For each setting, we train 3 models with different random seeds, and show the median error (solid lines), and the minimum and maximum errors (shaded region). (See Figures 6b, 6c in the main text for the corresponding plots on different-orthants and skewed-covariance.)

## B.4 Training variance

In Figure 11, we show the variance in error across training runs for the standard Transformer model (22.4M parameters). We plot the squared error for 3 models (with different random seeds) each for $d \in \{10, 20, 30, 40, 50\}$, trained to in-context learn linear functions. The error is quite concentrated in the standard setting as well as for most out-of-distribution prompts. In the different-orthants and skewed-covariance settings, we observe a high variance for higher dimensional problems ($d \geq 30$). However, in Section 6, we saw that the error in these settings usually decreases as we increase the model size. In the setting where we scale $x$, there is high variance even when $d = 10$.

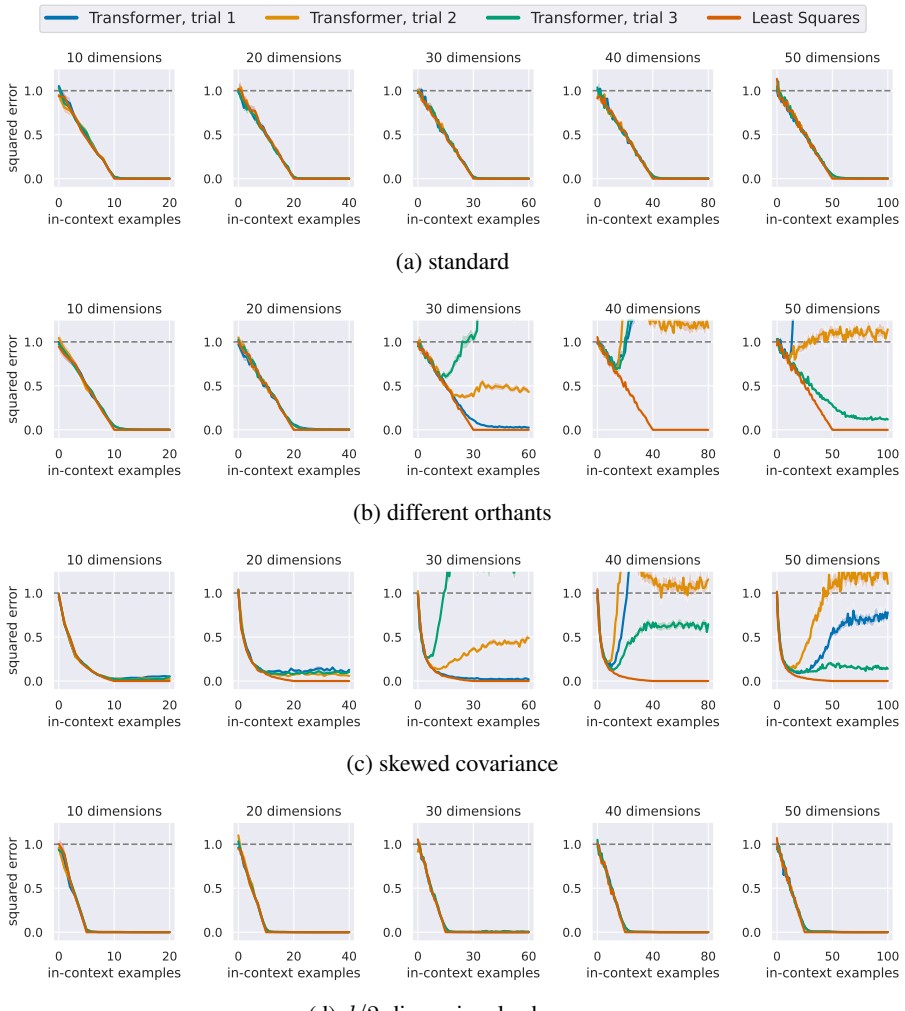

Figure 11: *Errors for models trained with different random seeds.* For each dimension, we train three models with different random seeds and show the corresponding error curves.

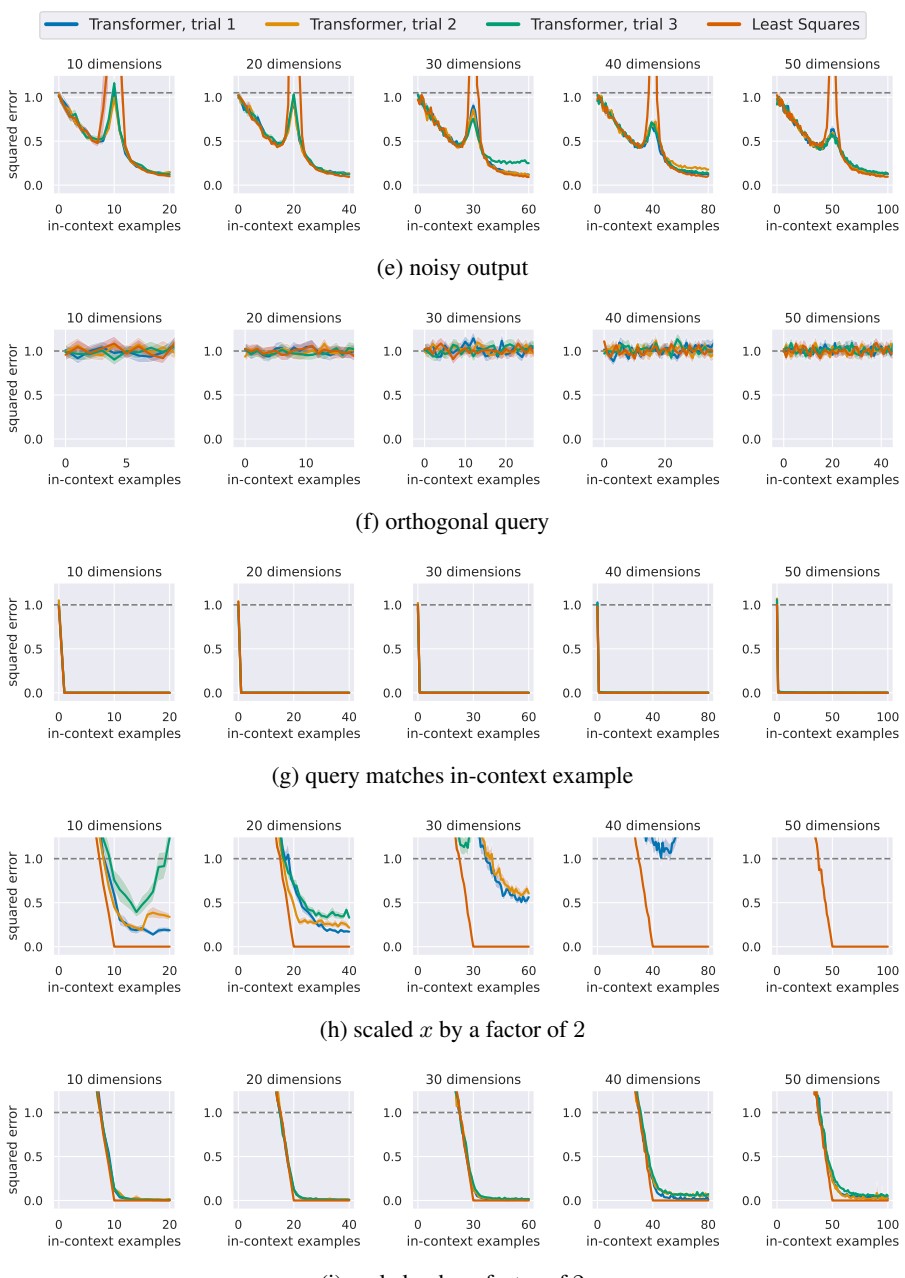

(e) noisy output

(f) orthogonal query

(g) query matches in-context example

(h) scaled $x$ by a factor of 2

(i) scaled $w$ by a factor of 2

Figure 11: (continued) *Errors for models trained with different random seeds.* For each dimension, we train three models with different random seeds and show the corresponding error curves.

## B.5 Curriculum

In Figure 12, we show the training loss of the Transformer model trained to in-context learn linear functions, with and without a curriculum. Specifically, given a random training prompt sequence $(x_1, f(x_1), x_2, f(x_2), \ldots, x_{k_{\mathrm{cur}}}, f(x_{k_{\mathrm{cur}}}))$, let $\hat{y}_i$ be the model's prediction for the $i^{\mathrm{th}}$ input (meant to approximate $f(x_i)$). For each such prompt, we consider the loss given by the normalized squared error averaged over all prompt prefixes

$$\frac{1}{k_{\mathrm{cur}}} \sum_{i=1}^{k_{\mathrm{cur}}} \frac{(\hat{y}_i - f(x_i))^2}{d}.$$

At each training step, we plot the loss averaged over a batch of $64$ random prompts. For training with curriculum, $k_{\mathrm{cur}}$ is gradually increased to $2d + 1$ as described in Section A.2. For training without curriculum $k_{\mathrm{cur}} = 2d + 1$ at all times.

Note that the loss often increases in the beginning as we train the model with curriculum. This is due to a sharp increase in the loss at steps where we increase the effective dimensionality ($d_{\mathrm{cur}}$) of prompt inputs ($x_i$). There are two reasons for this increase: (i) variance of the target output ($f(x_i) = w^\top x_i$) increases, so even the optimum loss is larger, (ii) the model performance is worse for the prompt inputs with increased effective dimension. After each such step where we increment $d_{\mathrm{cur}}$, the loss starts to decrease again until the next increment. The overall trend in the loss looks upward when the sharp increase dominates the decrease that follows. Some observations worth highlighting are as follows.

**Curriculum drastically speed-ups training.** For functions in 20 or more dimensions, curriculum allows us to train a low-error model often 4 times faster. Moreover, training without curriculum does not always succeed within our training budget (500k steps), e.g., for one run with $d = 30$ and *all* runs with $d = 50$, the loss does not decrease at all without curriculum.

**Initial lull without curriculum.** For training without curriculum, we observe that the loss does not decrease for relatively a long period in the beginning, and starts to decrease sharply thereafter. There is a large variance in the length of this period for any fixed dimension, and the average length seems to increase with dimension. This period is almost non-existent for smaller dimensions (e.g., see the plot for $d = 10$), and therefore we do not observe such a period while training with curriculum where we start training with inputs lying in a 5 dimensional subspace.

**Curriculum does not affect final performance significantly.** For our core setting ($d = 20$), where we are able to train the model to low error even without curriculum, we do not observe any qualitative differences in the error in most cases (both with and without distribution shifts). One exception is the case with skewed covariance, where the model trained without curriculum seems to do slightly better. We plot the error curves for the standard, different orthants and skewed covariance cases in Figure 13.

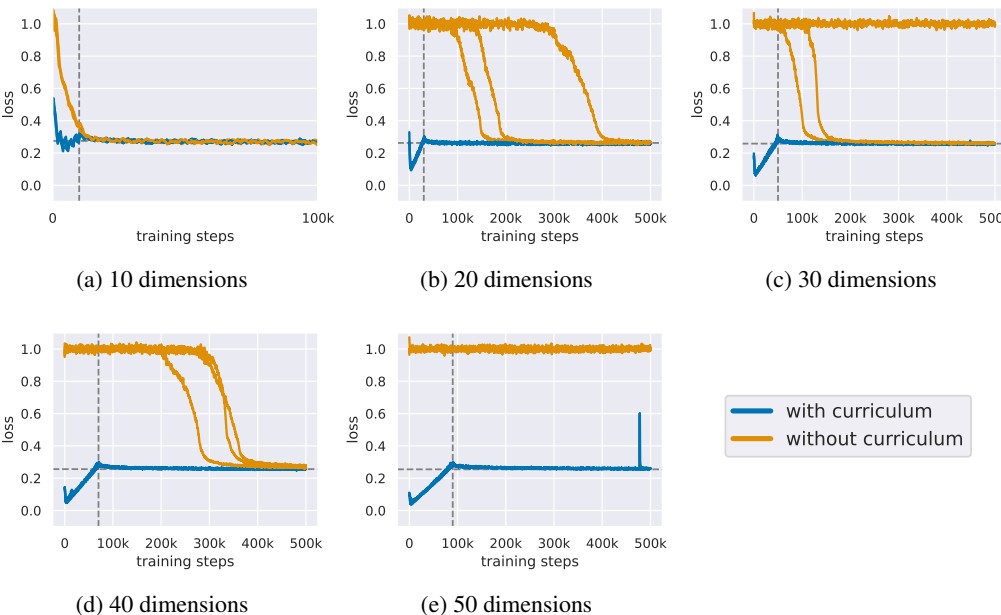

(a) 10 dimensions       (b) 20 dimensions       (c) 30 dimensions

(d) 40 dimensions       (e) 50 dimensions

Figure 12: *Loss progression during training with and without curriculum.* For each dimension, we show the loss progression with 3 random seeds each for training with and without curriculum. The vertical dashed line shows the point at which the effective dimension of prompt inputs $d_{\text{cur}}$ reaches the actual dimension $d$, after which training with and without curriculum have the same prompt distribution. The horizontal dashed line shows the optimum expected loss. There is a drastic speedup in training with curriculum. Without curriculum, there is an initial relatively long period where the loss does not decrease. For each dimension, there is a large variance in the length of this period, and the average length seems to increase with dimension.

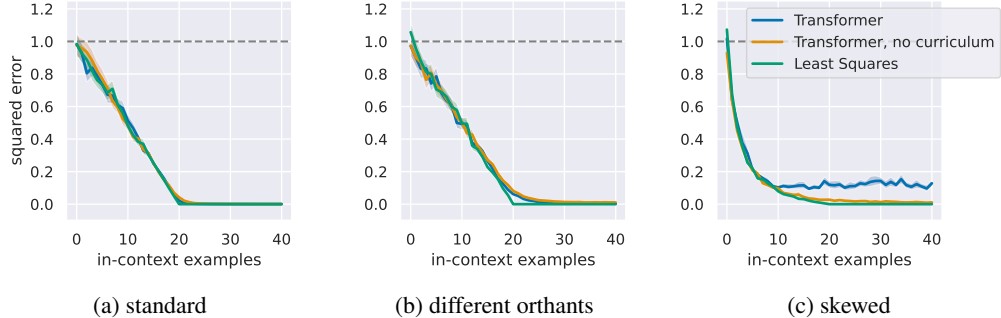

(a) standard       (b) different orthants       (c) skewed

Figure 13: *In-context learning performance for models trained with and without curriculum.* We show the performance for models trained with and without curriculum for in-context learning linear functions ($d = 20$). We did not observe any major qualitative difference in performance between the two settings in most cases. One exception is the case with skewed covariance where the model trained without curriculum does better.

## B.6 Effect of number of distinct prompts/functions seen during training

Here, we investigate the effect of amount of training data required for in-context learning linear functions.

First, we consider the effect of number of distinct prompts encountered during training. For this, we create a set $S_p$ of $n_p$ randomly generated prompts, where each prompt in $S_p$ is generated by sampling a weight vector and prompt inputs from $N(0, I_d)$. We generate random prompts during training by sampling uniformly from this set. As before, we train the model for $500k$ steps with a batch size of $64$. We observe (see Figure 14) that a model trained with $n_p = 100k$ is able to achieve non-trivial error and a model trained with $n_p = 1M$ achieves error close to that of the unrestricted model (trained with $32M$ distinct prompts). Recall that with curriculum learning, we zero out some of the coordinates of prompt inputs in the beginning of training, which will increase the total number of prompts the model sees during training. Therefore we do not use curriculum learning for this study to avoid inflating the number of distinct prompts seen during training.

Second, we consider the effect of number of distinct weight vectors (equivalently, distinct functions) encountered during training. For this, we create a set $S_w$ of $n_w$ weight vectors where each weight $w$ is drawn i.i.d. from $N(0, I_d)$. To generate a training prompt, $(x_1, w^\top x_1, \ldots, x_k, w^\top x_k)$, we draw prompt inputs ($x_i$s) i.i.d. from $N(0, I_d)$ as in the unrestricted setting, and sample $w$ uniformly at random from $S_w$. Thus while we sample from a finite set of weight vectors, we sample fresh inputs at each step. As before, we train the model for $500k$ steps with a batch size of $64$. Here, we observe (see Figure 14) that the model trained with as few as $10k$ distinct weight vectors achieves error close to the unrestricted model (trained with $32M$ distinct functions). We use curriculum learning for this study as in our standard setting. Recall that with curriculum learning, we only zero out some coordinates of prompt inputs in the beginning, so this does not change the number of distinct weight vectors seen by the model during training.

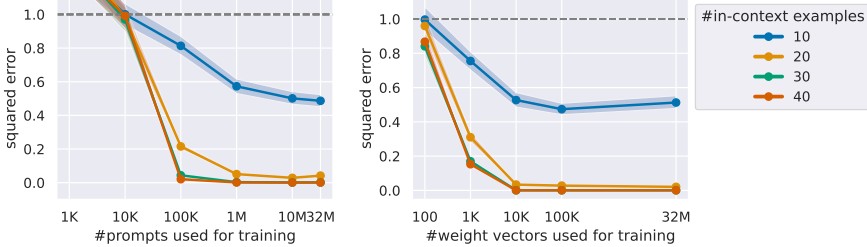

Figure 14: *Effect of number of distinct prompts/functions seen during training.* We plot the squared error for models trained to in-context linear functions, as we increase the number of distinct prompts and distinct weight vectors (equivalently, distinct functions) seen during training. (Note that 32M corresponds to the unrestricted model where we sample fresh prompts at each training step.) The models are able to achieve error close to that of the unrestricted model with $1M$ distinct prompts or $10k$ distinct weight vectors.

## B.7 Can memorization explain model performance?

Can memorization of training prompts or functions seen during training explain the model performance? Consider the setting of linear functions from Section 3. Note that the probability of the model encountering a training prompt similar to the one used for testing is astronomically low—the prompt inputs alone lie in a 800-dimensional space when predicting with $2d$ in-context examples ($d = 20$). Moreover, even considering the possibility that the model encountered a similar *weight vector* during training cannot explain its performance. That is, the model encounters 32 million random weight vectors during training and even using the best of these vectors would lead to an expected error of around 0.2 (computed empirically, details below). However, the model is able to achieve an error of less than 0.001 for a prompt with $2d$ in-context examples. Further, in Section 6, we show that the model is able to obtain a similar error even when trained on prompts generated using only $10,000$ distinct weight vectors, in which case the best weight vector seen during training would yield an even worse error of around 0.5. Thus, the model cannot be relying on memorization of training prompts or weight vectors, and instead encodes an algorithm capable of in-context learning linear functions that are very different from those seen during training.

**Error for the algorithm memorizing the weight vectors.** Let $S_w$ be the set of weight vectors used to generate training prompts. At inference time, given a prompt with in-context examples generated using a weight vector $w_\star$, suppose the model is somehow able to find the best weight vector $\hat{w}$ in $S_w$ minimizing the normalized squared error on query inputs:

$$\hat{w} = \arg\min_{w \in S_w} \mathbb{E}_{x_{\text{query}} \sim N(0, I_d)} \left[ \frac{(w^\top x_{\text{query}} - w_\star^\top x_{\text{query}})^2}{d} \right]$$
$$= \arg\min_{w \in S_w} \frac{\|w - w*\|_2^2}{d}$$

Taking expectation over the weight vector $w_\star$, we get the expected normalized squared error of the model (with respect to randomly drawn in-context examples and query inputs):

$$\mathbb{E}_{w_\star \sim N(0, I_d)} \left[ \min_{w \in S_w} \frac{\|w - w_\star\|_2^2}{d} \right].$$

To empirically estimate this quantity, we sample $n_w$ weight vectors from $N(0, I_d)$ (with $d = 20$) that form the set $S_w$, and 500 weight vectors from $N(0, I_d)$ to estimate the outer expectation. We do this 20 times, freshly sampling the 500 weight vectors and the vectors comprising $S_w$ each time, and compute the mean of the 20 estimates obtained. When $n_w = 32M$ (number of weight vectors encountered in our standard training setup), we get a mean of 0.216 (standard deviation 0.004). However, our model is able to achieve an expected error of less than 0.001 for prompts with $2d$ in-context examples. Similarly, when $n_w = 10,000$, we get a mean of 0.505 (standard deviation 0.006), while a model trained on prompts generated using $10,000$ distinct weight vectors is able to achieve a much smaller error (see Figure 14).