# OpenReview forum: "What Can Transformers Learn In-Context? A Case Study of Simple Function Classes"
_NeurIPS.cc/2022/Conference — NeurIPS 2022 Accept_

### Official Review · Reviewer_NrcW · 2022-07-10

**Rating:** 7
**Confidence:** 5
**Soundness:** 4 excellent
**Presentation:** 4 excellent
**Contribution:** 3 good

**Summary:**

The paper studies in-context learning in transformers by training models to predict the output of a linear function given a few input/output examples presented in context. The main findings are that transformers trained in this way are able to match the performance of the optimal estimator in various settings, and show some ability to generalize out of distribution. Surprisingly, models trained on sparse linear functions have an in-context performance that matches Lasso. Also surprisingly, models trained on the output of 2-layer ReLU networks perform reasonably well on linear functions. Finally, the authors find that curriculum learning (starting with easier examples and progressing to harder ones) greatly speeds up training.

**Questions:**

It's interesting to explore how models are able to achieve in-context learning of linear and sparse linear functions. For example, can one recover the linear function's weights from the transformer's activations? And does the transformer implement some version of Lasso, or does it work in a different way? I realize this is out of scope for the present work. I'm only proposing these questions in case the authors have more to share on this.

In Figure 2 "trivial error" is introduced but is not defined. I believe it is defined to be the zero estimator only in Figure 5.

**Limitations:**

This work has no obvious societal impact.

**Strengths And Weaknesses:**

The paper studies in-context learning in a well-scoped and well controlled setting. The setup and results are presented clearly, and the experiments are designed to produce sharp results with clear baselines and conclusions. I found several of the results surprising and inspiring for future investigation. The experiments on OOD generalization were especially interesting. Understanding in-context learning in transformers is an important problem, and this paper is a good step in that direction.

In terms of weaknesses, the settings studied in this work are relatively simple, and it is still hard to see how they connect with examples of in-context learning using natural data.

---

> ### Author Response · Authors · 2022-08-02
> **Author response**
>
> We thank the reviewer for their positive comments and feedback.
>
> The main weakness pointed out is the connection of our work to the setting of natural data. From that perspective, one pertinent question regarding the in-context learning capabilities of language models is to how they leverage the information present in the in-context examples (e.g., see [Min et al. 2022](https://arxiv.org/abs/2202.12837)). Our results demonstrate that, in principle, Transformers can encode complex learning algorithms that utilize in-context examples in a far-from-trivial manner. Moreover, these Transformers can be trained efficiently without any changes to their architecture, with standard optimization procedures. Whether such non-trivial learning behavior exists in large language models is still open, but we believe that our work takes a step towards formalizing and investigating this question.
>
> ### Questions
>
> > [would be] interesting to explore how models are able to achieve in-context learning of linear and sparse linear functions
>
> We do agree with the reviewer that understanding this capability mechanistically is an interesting future research direction that we, unfortunately, were not able to explore within the scope of this work.
>
> > In Figure 2 "trivial error" is introduced but is not defined.
>
> Yes, this corresponds to the zero estimator. Thank you for pointing this out. We will clarify this in the revised paper.

---

### Official Review · Reviewer_oarv · 2022-07-11

**Rating:** 8
**Confidence:** 4
**Soundness:** 4 excellent
**Presentation:** 4 excellent
**Contribution:** 4 excellent

**Summary:**

This paper tries to study the in-context learnability of function classes in transformers. While the existing research is actively exploring this phenomenon in language models, this work presents a novel analysis of the phenomenon using function classes (in lieu of tasks) as the objects to be learnt. The paper shows that transformers can learn linear functions with high precision (obtaining almost the same loss as the least squares estimator) and that learning more complex  function classes (e.g. 2 layer neural nets) entails the in-context learning of simpler function classes (e.g. linear). The paper also shows that (similar to many language tasks in large-scale language models), learning function classes could also be classified as an "emergent" behavior during training. Further, the paper also analyze model behavior under distribution shift and how that in-context learning of function classes is surprisingly robust to two common forms of distribution shift. Altogether, this analysis is a solid contribution to the literature, with simple, easy to follow experiments that can shed more light into the mechanics of in-context learning.

**Questions:**

1. In section 5 (lines 229-232), does the model trained to learn the 2-layer neural net also learn sparse linear functions?
2. For a model with a given capacity, how does the data size change in-context learning performance?
3. The paper shies away from analyzing the interaction of training data and model capacity. Does the in-context learning performance depend on effective model complexity and not just on model capacity (as characterized by number of parameters)? Any comments on incorporating training data size into the analysis?
4. Unless I missed something, is there an experiment where model training is done a mixture of function classes? How sensitively do the the results depend on the mixture ratios?


**Limitations:**

The authors do not include a section on limitations and potential negative societal impacts of the work. I urge the authors to add a limitations section, explaining any factors (e.g. training data size, effective model complexity, etc.) that could further complicate the analysis.


**Strengths And Weaknesses:**

The strengths of the paper are:

1. The paper is well written, with a clearly defined (and well motivated) experimental protocol. This simple setup could encourage further experiments.
2. The empirical results are intriguing and will be of interest to the broader ML research community, given transformer has become such an indispensable architecture.
3. To my knowledge this is the first exposition of "emergent" in-context learning in function classes as well as curriculum learning of function classes.

The paper has the following weaknesses:

1. There is no analysis of the size of training data that is required for successful in-context learning. A surprising omission in the experiments is the lack of a plot of data size against in-context learning performance. I think this is an important component that is missing.
2. The paper does not shed any light on the in-context learning phenomenon for language tasks, and one of the missing pieces is it's lack of analysis on training data size, along with training on a mixture of function classes.

---

> ### Author Response · Authors · 2022-08-02
> **Author response**
>
> We thank the reviewer for their positive feedback and thoughtful suggestions.
>
> The main points brought up by the reviewer concern the lack of training data size analysis and the connection of our work to the setting of natural language. We respond to each of these points below, and then respond to the rest of the questions.
>
> **[Impact of training data size]** We include an analysis of the number of distinct functions used to train the model in Supplementary Section B.6 (Figure 13). We also include an additional analysis below where we sample a fixed set of prompts and train on these alone (will add these results to the updated manuscript). In general, we find that the amount of training data required is relatively small: non-trivial in-context learning is possible with 1k distinct linear functions or 10k random prompts, whereas the error drops close to that of the unrestricted model for 10k linear functions or 100k prompts.
> | Training prompts | 20 examples | 30 examples | 40 examples
> | --- | ----------- |  -------- | --------
> | 1k    | 1.28 |  1.30 | 1.24
> | 10k    | 0.19 |  0.045 | 0.022
> | 100k    | 0.095 | 0.0049  | 0.0019
> | 1m    |  0.051 |  0.0027 | 0.0021
> | 10m    | 0.025 |  0.0009  | 0.0006
> | 32m (standard)   | 0.021 |  0.0008 | 0.0006
> |
>
> **[Connection to language modeling]** One pertinent question regarding the in-context learning capabilities of language models is to how they leverage the information present in the in-context examples (e.g., see [Min et al. 2022](https://arxiv.org/abs/2202.12837)). Our results demonstrate that, in principle, Transformers can encode complex learning algorithms that utilize in-context examples in a far-from-trivial manner. In fact, this is the case for standard Transformer architectures trained with standard optimization tools. Whether such non-trivial in-context learning behavior exists in large language models is still open, but we believe that our work takes a step towards formalizing and investigating this question.
>
> ### Questions
>
> > is there an experiment where model training is done a mixture of function classes?
>
> We decided to focus on a clean setting where we only train one model per function class. Nevertheless, we agree that training on mixtures of functions is a valuable direction for future investigation.
>
> > does the model trained to learn the 2-layer neural net also learn sparse linear functions
>
> We did not try this but we don't expect the model to be able to take advantage of sparsity, as it is not trained on any sparse inputs. We expect its performance on sparse linear functions to be worse than that of least squares (which does not take advantage of sparsity), as in the case of non-sparse linear functions.
>
> > The authors do not include a section on limitations and potential negative societal impacts of the work
>
> From our perspective, the main limitations of our work are: (a) we do not fully understand the learning algorithms implemented by the Transformers, (b) we cannot draw direct conclusions about language models since we study a synthetic setting (even though our results are still relevant as discussed above). While we mention some of these limitations in our discussion of future work, we are happy to add an explicit section in the updated manuscript. We do not perceive any tangible negative social impact that stems directly from our work.

---

> > ### Comment · Reviewer_oarv · 2022-08-08
> > **Interesting Additional Results**
> >
> > I appreciate the additional experiment added above, in my opinion, it is quite surprising (?) that in-context-learning is working non-trivially with such small training size.

---

### Official Review · Reviewer_aHn3 · 2022-07-11

**Rating:** 8
**Confidence:** 4
**Soundness:** 4 excellent
**Presentation:** 4 excellent
**Contribution:** 4 excellent

**Summary:**

**Goal:** Investigate to what extent decoder-only transformer models are capable to in-context learn functions from simple functions classes, like linear functions, sparse linear functions and shallow neural networks. If the answer is positive, understand the training conditions that lead to such in-context learning capabilities.

**Main results:**
* Transformers are capable of in-context learning all of the simple function classes tested in the paper to a surprising extend. The mean squared error values computed using the transformers often match the best-case known algorithms (such as min-norm linear regression and LASSO).  (The extent to which this is true is quite striking)
* Carefully designed experiments show that this capability isn't explained by simple memorization of training examples. Rather, the transformers seem to learn in-context learning strategies that generalize to nontrivial distribution shifts.
  * Distribution shifts in the in-context exemplars, as well as between in-context and query examples mostly leave the performance unchanged.
  * In the case of learning linear functions, even the gradient of the learned transformer matches that of the min-norm linear regression solution. This is also a very strong and surprising finding.

====================

**Post rebuttal update**

I thank the authors for their response. I still think that this is a strong submission that deserves a place in NeurIPS2022, hence maintain my score.

**Questions:**

* Regarding curriculum learning: Do you think one can do away with curriculum learning by using a fixed dataset that contains a mixture of examples of different difficulties? Establishing this could be useful for facilitating future research in this topic, as one does not need to deal with the additional subtleties regarding a curriculum learning setup.
* Inputs to the transformer: I don't quite get the description between lines 107 and 109. How exactly are the exemplars processed before being fed to the first transformer layer?
* Consistency of results: Does training work all the time, or do you need to train a number of times before you come across something that displays the impressive in-context learning results?
* Figure 3 (b): Is this plot averaged over multiple prompts? (This is a very interesting finding!)


Nitpick: Typo at line 335: "when being trained" instead of "when being training"
Another nitpick: In line 307, you write that you study "what classes of functions can a model learn in-context". The current set of results don't seem to address this exact framing, as answering that question would require experiments with many more types of functions. Perhaps rephrasing this would be helpful?

**Limitations:**

I don't wish to report any glaring limitations beyond what has been listed as weaknesses above.

**Strengths And Weaknesses:**

STRENGTHS
* **Comprehensive experiments:** The paper is full of clever experiments that strongly support the claims made in the paper. A number of these experiments yield nontrivial and (in my opinion) unexpected and surprising results.
* **Sound and convincing:** The claims made in the paper are sound and convincing (assuming the experiments were run correctly).
  * Some of the results look "too good to be true" (i.e. to the extent transformers match the performance of optimal algorithms like LASSO and min-norm linear regression), though the experiments look carefully done and I don't currently have any suspicions as to their validity.
* **Well scoped and crisp problem formulation and analysis** The paper is very well scoped, and the goal and the contributions are very clear.



WEAKNESSES
* **Lack of discussion on types of in-context learning** There are multiple forms of in-context learning that can perhaps be studied separately (Olsson et. al. [31] discuss this briefly). This paper tackles only one of those - it would be better if the authors disambiguated this earlier in the paper.
* **More distribution shifts:** Perhaps one or two more distributions shifts in addition to the skewed Gaussian distribution could be helpful.
(The paper is very well-scoped, and (in my opinion) doesn't have that many glaring limitations within that scope. Some of what's below can be regarded as recommendations to make the paper better instead of weaknesses)
* **Visualizing Sampled 2-layer neural network functions:** It'd be useful to provide some kind of visualization to make sure that the approximated neural network functions are non-trivial in the domain they're evaluated at. For example, all of the sampled neural nets could be roughly linear around the domain most of the samples fall, which'd make the result a bit less interesting (though still cool).
* **Lack of mechanistic understanding:** After the authors make a very compelling case that transformers do indeed learn generic in-context learners, perhaps the authors could have spent some time analyzing how a GPT-2 like architecture could be achieving this mechanistically.

---

> ### Author Response · Authors · 2022-08-02
> **Author response (part 1/2)**
>
> We thank the reviewer for their thoughtful feedback and positive comments.
>
> The main points raised by the reviewer pertain to the definition of in-context learning and potential additional experimental analysis of our results. We first address these main points and then respond to individual questions in the next part of our reply.
>
> **[Other types of in-context learning]** We follow the standard practice of using the term “in-context learning” to refer to the ability of a model to condition on a prompt sequence consisting of in-context examples from a task along with a new query input, and generate the corresponding output (e.g., see [Brown et al. 2020](https://arxiv.org/abs/2005.14165) who coined this terminology,  [Xie et al. 2021](https://arxiv.org/abs/2111.02080), and [Min et al. 2022](https://arxiv.org/abs/2202.12837)). Nevertheless, we are aware of other works ([Elhage et al. 2021](https://transformer-circuits.pub/2021/framework/index.html) and [Olsson et al. 2022](https://transformer-circuits.pub/2022/in-context-learning-and-induction-heads/index.html)) that consider a more general notion, where "in-context learning" refers to any model behavior that utilizes information in the prompt to make predictions that improve with prompt size. We will clarify this point in the revised manuscript.
>
> **[Studying more types of distribution shift]** In terms of changing the distribution of prompt inputs, in addition to a skewed Gaussian, we also consider inputs sampled from a low-dimensional subspace (Supplementary Figure 8a) as well as inputs sampled from a single orthant (Figure 4c). We found these shifts to be relatively natural and informative, but we agree that there are (many) more things one can try.
>
> **[Are neural networks roughly linear in the regime we study?]** In Figure 5b we can see that a linear regression baseline (least squares) does not approximate the predictions of the neural network as well. We can thus conclude that the neural networks cannot be too close to linear. Moreover, we also consider the class of decision trees of depth 4: each internal node branches based on the sign of a randomly-chosen coordinate and each leaf corresponds to a randomly sampled scalar value (this is a highly nonlinear class of functions). We find that Transformers can be trained to in-context learn this function class as well. We will add this to the updated manuscript.
>
> | Model | 25 examples | 50 examples | 100 examples
> | --- | ----------- |  -------- | --------
> | 3-Nearest Neighbors    | 1.13 | 1.11  | 1.01  |
> | Greedy Tree Learning    | 1.68 | 1.42 | 1.03 |
> | XGBoost    | 1.24 |  0.94 | 0.73 |
> | Transformer    | **0.59** | **0.28** | **0.12** |
> (the trivial zero estimator achieves an error of 1.0)
>
> **[Mechanistic understanding of Transformers]** We agree that digging into the internals of these models is a valuable future research direction. As we note in our discussion section, not only would this shed light into the internal workings of Transformers but also potentially help us discover learning algorithms for challenging problems.

---

> > ### Author Response · Authors · 2022-08-02
> > **Author response (part 2/2)**
> >
> > We now respond to individual questions raised by the reviewer.
> >
> > > Do you think one can do away with curriculum learning by using a fixed dataset that contains a mixture of examples of different difficulties?
> >
> > We did not perform such experiments but in principle it might be possible. Nevertheless, this would still require tuning the ratio of function classes to ensure that both: (a) the model can start training by consistently encountering easy functions, (b) the model encounters enough samples of the target function class (i.e., maximum dimensionality) to achieve good performance there.
> >
> > > How exactly are the exemplars processed before being fed to the first transformer layer?
> >
> > The Transformer architecture takes as input a sequence of vectors in its embedding space and predicts the next vector in the sequence within the same embedding space (in language modeling these vectors correspond to input tokens). We apply this architecture to our prompt format of $(x_1, f(x_1),\ldots, x_{k+1}, f(x_{k+1}))$ as follows. We map each prompt output $f(x_i)$ to the same dimension as prompt inputs $x_i$ by appending zeros, and map the prompt inputs and outputs into the latent embedding space of the Transformer through a (learnable) linear transformation. We then use another (learnable) linear transformation to map the vector produced by the model to a scalar. Note that the Transformer architecture allows us to compute the prediction for all prompt prefixes in a single forward pass. We will edit our manuscript to clarify any confusion.
> >
> > > Consistency of results: Does training work all the time, [...]?
> >
> > We present the results of three random trials in Supplementary Figure 10 (the results in the main paper correspond to the first of these trials). In general, when considering the standard setting where prompts follow the same distribution during training and testing, all models learn consistently and achieve low loss (even for the harder task of linear regression in 50 dimensions). We do however see some variance in certain out-of-distribution prompts in higher dimensions (e.g., different orthants for dimension $\geq 30$ in Figure 10b) but it might be possible to reduce this by training models of larger capacity, as indicated by our capacity experiments (Fig 6).
> >
> > > Figure 3 (b): Is this plot averaged over multiple prompts?
> >
> > The plot is averaged over 1280 prompts with 95% confidence intervals using 1000 bootstrap trails. We will mention this in the revised paper.
> >
> > > Nitpick [...]
> >
> > We thank the reviewer for pointing these out. We will update these in the revised manuscript.

---

### Official Review · Reviewer_VCf1 · 2022-07-12

**Rating:** 6
**Confidence:** 4
**Soundness:** 3 good
**Presentation:** 4 excellent
**Contribution:** 3 good

**Summary:**

This paper proposes to study the ability of transformers model to predict values of a function given an input via sample prompting. Overall, this is a nice study that sheds some light on the capability of transformers model to deduce what function the context refers to and execute/compute the value of that function given an input in the context.

**Questions:**

- For the input format x_1, f(x_1), x_2, f(x_2), .. if x_1 is a 20-dim vector, how is it presented to the model. This is a sequence of 20 numbers followed by 1 number for f(x_1, and so on?
- any comparison regarding the overall loss function proposed?

**Limitations:**

I think the paper can easily be expanded to include complex functions. 2-layer ReLU is more complex than linear but also not as interpretable. Extending to something like degree-m polynomial and to observe trends with respect to capacity / how much it takes to train / etc can shed more light into the ability of transformers model for this type of sequence modeling.

**Strengths And Weaknesses:**

Pros:

- Quite a thought-provoking paper studying transformers capabilities to learn explicit mathematical functions
- Nice analysis / experiments. Everything seems well designed given the scope

Cons:
- There's a lot of mentioning for 'in-context'. However, I do fear that this could be a bit misleading. All the model is doing is doing sequential modeling that given a series of inputs and the corresponding function values, the model predicts the function value of  the previous input. So in a way this can also be seen as sequence modeling.

---

> ### Author Response · Authors · 2022-08-02
> **Author response**
>
> We thank the reviewer for their encouraging feedback and positive comments.
>
> The main points raised by the reviewer pertain to the definition of in-context learning as well as expanding the analysis to include more complex functions. We address these points below and then respond to clarifying questions.
>
> **[Is in-context learning just "sequence modeling"?]** In principle yes, any task performed by auto-regressive models can be framed as sequence modeling. However, here we focus on a more specific notion of sequence modeling, corresponding to predicting on sequences that consist of examples from some tasks. These sequences have more structure and thus this behavior/capability has been studied separately in the literature under the name of "in-context learning" [[Brown et al. 2020](https://arxiv.org/abs/2005.14165), [Xie et al. 2021](https://arxiv.org/abs/2111.02080), [Min et al. 2022](https://arxiv.org/abs/2202.12837)].
>
> **[Studying more complex function classes]** We also performed experiments using depth-4 decision trees: each internal node branches based on the sign of a randomly-chosen coordinate and each leaf corresponds to a randomly sampled scalar value (this can be seen as a degree 4 polynomial over the sign of the input coordinates). We find that Transformers can also learn this class of functions, significantly outperforming the greedy learning algorithm and tree boosting (XGBoost). We will add this analysis to the final manuscript.
> | Model | 25 examples | 50 examples | 100 examples
> | --- | ----------- |  -------- | --------
> | 3-Nearest Neighbors    | 1.13 | 1.11  | 1.01  |
> | Greedy Tree Learning    | 1.68 | 1.42 | 1.03 |
> | XGBoost    | 1.24 |  0.94 | 0.73 |
> | Transformer    | **0.59** | **0.28** | **0.12** |
> (the error of the trivial zero estimator is 1.0)
>
> ### Questions
>
> > How is it [the input] presented to the model?
>
> The Transformer architecture takes as input a sequence of vectors in its embedding space and predicts the next vector in the sequence within the same embedding space (in language modeling these vectors correspond to input tokens). We apply this architecture to our prompt format of $(x_1, f(x_1),\ldots, x_{k+1}, f(x_{k+1}))$ as follows. We map each prompt output $f(x_i)$ to the same dimension as prompt inputs $x_i$ by appending zeros, and map the prompt inputs and outputs into the embedding space of the Transformer through a (learnable) linear transformation. We then use another (learnable) linear transformation to map the vector produced by the model to a scalar. Note that the Transformer architecture allows us to compute the prediction for all prompt prefixes in a single forward pass.  We will edit our manuscript to clarify any confusion.
>
> > any comparison regarding the overall loss function proposed?
>
> The squared loss is one of the most natural loss functions to consider for regression problems. We thus did not experiment with any alternatives.

---

### Meta-Review · Area_Chair_SRVX · 2022-08-25

**Recommendation:** Accept
**Confidence:** Certain

**Metareview:**

This paper demonstrates compellingly that transformers are able to in-context learn simple function classes (e.g., linear functions), to the extend that they can recover solutions from algorithms like LASSO. The experiments are well designed and executed, which lead to surprising and intriguing results. While the paper does not provide any explanation for why transformers exhibit such capabilities, it will spur both empirical and theoretical work studying how transformers learn algorithms from in-context examples. Congratulations on a nice work!

**Award:**

No

---

### Decision · Program_Chairs · 2022-09-14

Accept